# COHORT SQUEEZE: BEYOND A SINGLE COMMUNICATION ROUND PER COHORT IN CROSS-DEVICE FEDERATED LEARNING

## ABSTRACT

Virtually all federated learning (FL) methods, including FedAvg, operate in the following manner: i) an orchestrating server sends the current model parameters to a cohort of clients selected via certain rule, ii) these clients then independently perform a local training procedure (e.g., via SGD or Adam) using their own training data, and iii) the resulting models are shipped to the server for aggregation. This process is repeated until a model of suitable quality is found. A notable feature of these methods is that each cohort is involved in a single communication round with the server only. In this work we challenge this algorithmic design primitive and investigate whether it is possible to "squeeze more juice" out of each cohort than what is possible in a single communication round. Surprisingly, we find that this is indeed the case, and our approach leads to up to 74% reduction in the total communication cost needed to train a FL model in the cross-device setting. Our method is based on a novel variant of the stochastic proximal point method (SPPM-AS) which supports a large collection of client sampling procedures some of which lead to further gains when compared to classical client selection approaches.

## 1 INTRODUCTION

Federated Learning (FL) is increasingly recognized for its ability to enable collaborative training of a global model across heterogeneous clients, while preserving privacy (McMahan et al., 2016; 2017; Kairouz et al., 2019; Li et al., 2020a; Karimireddy et al., 2020b; Mishchenko et al., 2022b; Malinovsky et al., 2024; Yi et al., 2024). This approach is particularly noteworthy in cross-device FL, involving the coordination of millions of mobile devices by a central server for training purposes (Kairouz et al., 2019). This setting is characterized by intermittent connectivity and limited resources. Consequently, only a subset of client devices participates in each communication round. Typically, the server samples a batch of clients (referred to as a *cohort* in FL), and each selected client trains the model received from the server using its local data. Then, the server aggregates the results sent from the selected cohort. Another notable limitation of this approach is the constraint that prevents workers from storing states (operating in a stateless regime), thereby eliminating the possibility of employing variance reduction techniques. We will consider a reformulation of the cross-device objective that assumes a finite number of workers being selected with uniform probabilities. Given that, in practice, only a finite number of devices is considered, i.e. the following finite-sum objective is considered:

$$\min_{x \in \mathbb{R}^d} f(x) := \frac{1}{n} \sum_{i=1}^{n} f_i(x). \tag{1}$$

This reformulation aligns more closely with empirical observations and enhances understanding for illustrative purposes. The extension to the expectation form of the following theory can be found in Appendix F.4.

Current representative approaches in the cross-device setting include FedAvg and FedProx. In our work, we introduce a method by generalizing stochastic proximal point method with arbitray sampling and term as SPPM-AS. This new method is inspired by the stochastic proximal point method

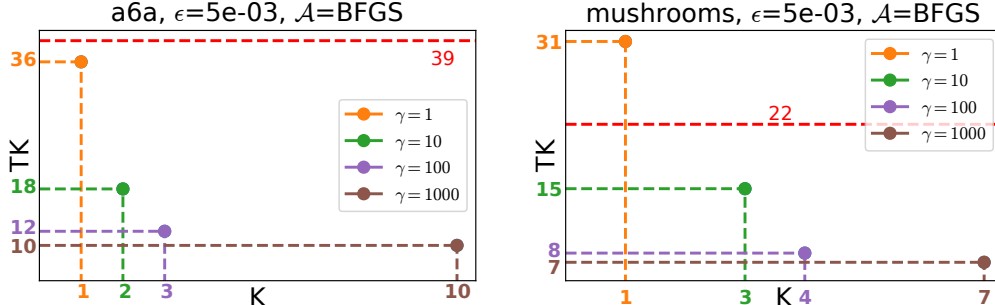

Figure 1: The total communication cost (defined as $TK$) with the number of local communication rounds $K$ needed to reach the target accuracy $\epsilon$ for the chosen cohort in each global iteration. The dashed red line depicts the communication cost of the FedAvg algorithm. Markers indicate the $TK$ value for different learning rates $\gamma$ of our algorithm SPPM-AS.

(SPPM), a technique notable for its ability to converge under arbitrarily large learning rates and its flexibility in incorporating various solvers to perform proximal steps. This adaptability makes SPPM highly suitable for cross-device FL (Li et al., 2020a; Yuan & Li, 2022; 2023; Khaled & Jin, 2023; Lin et al., 2024). Additionally, we introduce support for an arbitrary cohort sampling strategy, accompanied by a theoretical analysis. We present novel strategies that include support for client clustering, which demonstrate both theoretical and practical improvements.

Another interesting parameter that allows for control is the number of local communications. Two distinct types of communication, *global* and *local*, are considered. A *global* iteration is defined as a single round of communication between the server and all participating clients. On the other hand, *local* communication rounds are synchronizations that take place within a chosen cohort. Additionally, we introduce the concept of total communication cost, which includes both local and global communication iterations, to measure the overall efficiency of the communication process. The total communication cost naturally depends on several factors. These include the local algorithm used to calculate the prox, the global stepsize, and the sampling technique.

Previous results on cross-device settings consider only one local communication round for the selected cohort (Li et al., 2020b; Reddi et al., 2020; Li et al., 2020a; Wang et al., 2021a;b; Xu et al., 2021; Malinovsky et al., 2023; Jhunjhunwala et al., 2023; Sun et al., 2023; 2024). Our experimental findings reveal that *increasing the number of local communication rounds within a chosen cohort per global iteration can indeed lower the total communication cost needed to reach a desired global accuracy level*, which we denote as $\varepsilon$. Figure 1 illustrates the relationship between total communication costs and the number of local communication rounds. Assume that the cost of communication per round is 1 unit. $K$ represents the number of local communication rounds per global iteration for the selected cohort, while $T$ signifies the *minimum* number of global iterations needed to achieve the accuracy threshold $\epsilon$. Then, the total cost incurred by our method can be expressed as $TK$. For comparison, the dashed line in the figure shows the total cost for the FedAvg algorithm, which always sets $K$ to 1, directly equating the number of global iterations to total costs. Our results across various datasets identify the optimal $K$ for each learning rate to achieve $\epsilon$-accuracy. Figure 1 shows that adding more local communication rounds within each global iteration can lead to a significant reduction in the overall communication cost. For example, when the learning rate is set to 1000, the optimal cost is reached with 10 local communication rounds, making $K = 10$ a more efficient choice compared to a smaller number. On the other hand, at a lower learning rate of 100, the optimal cost of 12 is reached with $K = 3$. This pattern indicates that as we increase the number of local communication rounds, the total cost can be reduced, and the optimal number of local communication rounds tends to increase with higher learning rates.

Our key *contributions* are summarized as follows:

• We present and analyze SPPM-AS, a novel approach within the stochastic proximal point method framework tailored for cross-device federated learning, which supports arbitrary sampling strategies. Additionally, we provide an analysis of standard sampling techniques and introduce new techniques based on clustering approaches. These novel techniques are theoretically analyzed, offering a thorough comparison between different methods.

• Our numerical experiments, conducted on both convex logistic regression models and non-convex neural networks, demonstrate that the introduced framework enables fine-tuning of parameters to surpass existing state-of-the-art cross-device algorithms. Most notably, we found that increasing the number of local communication rounds within the selected cohort is an effective strategy for reducing the overall communication costs necessary to achieve a specified target accuracy threshold.

• We offer practical guidance on the proper selection of parameters for federated learning applications. Specifically, we examine the potential choices of solvers for proximal operations, considering both convex and non-convex optimization regimes. Our experiments compare first-order and second-order solvers to identify the most effective ones.

## 2 METHOD

In this section, we explore efficient stochastic proximal point methods with arbitrary sampling for cross-device FL to optimize the objective (1). Throughout the paper, we denote $[n] := \{1, \ldots, n\}$. Our approach builds on the following assumptions.

**Assumption 2.1.** Function $f_i : \mathbb{R}^d \to \mathbb{R}$ is differentiable for all samples $i \in [n]$.

This implies that the function $f$ is differentiable. The order of differentiation and summation can be interchanged due to the additive property of the gradient operator.

$$\nabla f(x) \stackrel{Eqn. \ (1)}{=} \nabla \left[ \frac{1}{n} \sum_{i=1}^n f_i(x) \right] = \frac{1}{n} \sum_{i=1}^n \nabla f_i(x).$$

**Assumption 2.2.** Function $f_i : \mathbb{R}^d \to \mathbb{R}$ is $\mu$-strongly convex for all samples $i \in [n]$, where $\mu > 0$. That is, $f_i(y) + \langle \nabla f_i(y), x - y \rangle + \frac{\mu}{2} \|x - y\|^2 \leq f_i(x)$, for all $x, y \in \mathbb{R}^d$.

This implies that $f$ is $\mu$-strongly convex and hence has a unique minimizer, which we denote by $x_\star$. We know that $\nabla f(x_\star) = 0$. Notably, we do *not* assume $f$ to be $L$-smooth.

### 2.1 SAMPLING DISTRIBUTION

Let $\mathcal{S}$ be a probability distribution over the $2^n$ subsets of $[n]$. Given a random set $S \sim \mathcal{S}$, we define

$$p_i := \text{Prob}(i \in S), \quad i \in [n].$$

We restrict our attention to proper and nonvacuous random sets.

**Assumption 2.3.** $\mathcal{S}$ is proper (i.e., $p_i > 0$ for all $i \in [n]$) and nonvacuous (i.e., $\text{Prob}(S = \emptyset) = 0$).

Let $C$ be the selected cohort. Given $\emptyset \neq C \subseteq [n]$ and $i \in [n]$, we define

$$v_i(C) := \begin{cases} \frac{1}{p_i} & i \in C \\ 0 & i \notin C \end{cases} \Rightarrow f_C(x) := \frac{1}{n} \sum_{i=1}^n v_i(C) f_i(x) = \sum_{i \in C} \frac{1}{np_i} f_i(x). \tag{2}$$

Note that $v_i(S)$ is a random variable and $f_S$ is a random function. By construction, $\mathrm{E}_{S \sim \mathcal{S}}[v_i(S)] = 1$ for all $i \in [n]$, and hence

$$\mathrm{E}_{S \sim \mathcal{S}}[f_S(x)] = \mathrm{E}_{S \sim \mathcal{S}}\left[ \frac{1}{n} \sum_{i=1}^n v_i(S) f_i(x) \right] = \frac{1}{n} \sum_{i=1}^n \mathrm{E}_{S \sim \mathcal{S}}[v_i(S)] f_i(x) = \frac{1}{n} \sum_{i=1}^n f_i(x) = f(x).$$

Therefore, the optimization problem in Equation (1) is equivalent to the stochastic optimization problem

$$\min_{x \in \mathbb{R}^d} \{ f(x) := \mathrm{E}_{S \sim \mathcal{S}}[f_S(x)] \}. \tag{3}$$

Further, if for each $C \subset [n]$ we let $p_C := \text{Prob}(S = C)$, then $f$ can be written in the equivalent form

$$f(x) = \mathbb{E}_{S \sim \mathcal{S}}[f_S(x)] = \sum_{C \subseteq [n]} p_C f_C(x) = \sum_{C \subseteq [n], p_C > 0} p_C f_C(x). \tag{4}$$

## 2.2 Core Algorithm

Applying SPPM (Khaled & Jin, 2023) to Equation (3), we arrive at stochastic proximal point method with arbitrary sampling (SPPM-AS, Algorithm 1):

$$x_{t+1} = \text{prox}_{\gamma f_{S_t}}(x_t),$$

where $S_t \sim \mathcal{S}$.

---

**Algorithm 1** Stochastic Proximal Point Method with Arbitrary Sampling (SPPM-AS)

1: **Input:** starting point $x^0 \in \mathbb{R}^d$, distribution $\mathcal{S}$ over the subsets of $[n]$, learning rate $\gamma > 0$
2: **for** $t = 0, 1, 2, \ldots$ **do**
3:     Sample $S_t \sim \mathcal{S}$
4:     $x_{t+1} = \text{prox}_{\gamma f_{S_t}}(x_t)$
5: **end for**

---

**Theorem 1** (Convergence of SPPM-AS). Let Assumption 2.1 (differentiability) and Assumption 2.2 (strong convexity) hold. Let $\mathcal{S}$ be a sampling satisfying Assumption 2.3, and define

$$\mu_{\text{AS}} := \min_{C \subseteq [n], p_C > 0} \sum_{i \in C} \frac{\mu_i}{n p_i}, \quad \sigma_{\star,\text{AS}}^2 := \sum_{C \subseteq [n], p_C > 0} p_C \left\| \nabla f_C(x_\star) \right\|^2. \tag{5}$$

Let $x_0 \in \mathbb{R}^d$ be an arbitrary starting point. Then for any $t \geq 0$ and any $\gamma > 0$, the iterates of SPPM-AS (Algorithm 1) satisfy

$$\text{E}\left[ \|x_t - x_\star\|^2 \right] \leq \left( \frac{1}{1 + \gamma \mu_{\text{AS}}} \right)^{2t} \|x_0 - x_\star\|^2 + \frac{\gamma \sigma_{\star,\text{AS}}^2}{\gamma \mu_{\text{AS}}^2 + 2\mu_{\text{AS}}}.$$

**Theorem interpretation.** In the theorem presented above, there are two main terms: $\left( 1/(1+\gamma\mu_{\text{AS}}) \right)^{2t}$ and $\gamma \sigma_{\star,\text{AS}}^2 / (\gamma \mu_{\text{AS}}^2 + 2\mu_{\text{AS}})$, which define the convergence speed and neighborhood, respectively. Additionally, there are three hyperparameters to control the behavior: $\gamma$ (the global learning rate), AS (the sampling type), and $T$ (the number of global iterations). In the following paragraphs, we will explore special cases to provide a clear intuition of how the SPPM-AS theory works.

**Interpolation regime.** Consider the interpolation regime, characterized by $\sigma_{\star,\text{AS}}^2 = 0$. Since we can use arbitrarily large $\gamma > 0$, we obtain an arbitrarily fast convergence rate. Indeed, $\left( 1/(1+\gamma\mu_{\text{AS}}) \right)^{2t}$ can be made arbitrarily small for any fixed $t \geq 1$, even $t = 1$, by choosing $\gamma$ large enough. However, this is not surprising, since now $f$ and all functions $f_\xi$ share a single minimizer, $x_\star$, and hence it is possible to find it by sampling a small batch of functions even a single function $f_\xi$, and minimizing it, which is what the prox does, as long as $\gamma$ is large enough.

**A single step travels far.** Observe that for $\gamma = 1/\mu_{\text{AS}}$, we have $\gamma \sigma_{\star,\text{AS}}^2 / (\gamma \mu_{\text{AS}}^2 + 2\mu_{\text{AS}}) = \sigma_{\star,\text{AS}}^2 / 3\mu_{\text{AS}}^2$. In fact, the convergence neighborhood $\gamma \sigma_{\star,\text{AS}}^2 / (\gamma \mu_{\text{AS}}^2 + 2\mu_{\text{AS}})$ is bounded above by three times this quantity irrespective of the choice of the stepsize. Indeed, $\frac{\gamma \sigma_{\star,\text{AS}}^2}{\gamma \mu_{\text{AS}}^2 + 2\mu_{\text{AS}}} \leq \min \left\{ \frac{\sigma_{\star,\text{AS}}^2}{\mu_{\text{AS}}^2}, \frac{\gamma \sigma_{\star,\text{AS}}^2}{\mu_{\text{AS}}} \right\} \leq \frac{\sigma_{\star,\text{AS}}^2}{\mu_{\text{AS}}^2}$. That means that no matter how far the starting point $x_0$ is from the optimal solution $x_\star$, if we choose the stepsize $\gamma$ to be large enough, then we can get a decent-quality solution after a single iteration of SPPM-AS already! Indeed, if we choose $\gamma$ large enough so that $\left( 1/1+\gamma\mu_{\text{AS}} \right)^2 \|x_0 - x_\star\|^2 \leq \delta$, where $\delta > 0$ is chosen arbitrarily, then for $t = 1$ we get $\mathbb{E}\left[ \|x_1 - x_\star\|^2 \right] \leq \delta + \sigma_{\star,\text{AS}}^2 / \mu_{\text{AS}}^2$.

**Iteration complexity.** We have seen above that an accuracy arbitrarily close to (but not reaching) $\sigma_{\star,\text{AS}}^2 / \mu_{\text{AS}}^2$ can be achieved via a single step of the method, provided that the stepsize $\gamma$ is large enough. Assume now that we aim for $\epsilon$ accuracy, where $\epsilon \leq \sigma_{\star,\text{AS}}^2 / \mu_{\text{AS}}^2$. We can show that with the stepsize $\gamma = \varepsilon \mu_{\text{AS}} / \sigma_{\star,\text{AS}}^2$, we get $\text{E}\left[ \|x_t - x_\star\|^2 \right] \leq \varepsilon$ provided that $t \geq \left( \frac{\sigma_{\star,\text{AS}}^2}{2\varepsilon \mu_{\text{AS}}^2} + \frac{1}{2} \right) \log \left( \frac{2\|x_0 - x_\star\|^2}{\varepsilon} \right)$. We provide the proof in Appendix F.5. To ensure thoroughness, we present in Appendix F.9 the lemma of the inexact formulation for SPPM-AS, which offers greater practicality for empirical experimentation. Further insights are provided in the subsequent experimental section.

**General Framework.** By allowing the freedom to choose arbitrary algorithms for solving the proximal operator, one can see that SPPM-AS generalizes renowned methods such as FedProx (Li et al., 2020a) and FedAvg (McMahan et al., 2016). In doing so, we are able to adapt our theoretical framework to prove the convergence rate of FedProx using a minimalistic set of assumptions, specifically Assumption 2.1 and Assumption 2.2. A more detailed overview of these generalization properties is provided in Appendix B.4.

## 2.3 ARBITRARY SAMPLING EXAMPLES

Details on simple Full Sampling (FS) and Nonuniform Sampling (NS) are provided in Appendix B.2. In this section, we focus more intently on the sampling strategies that are of particular interest to us.

**Nice Sampling (NICE).** Choose $\tau \in [n]$ and let $S$ be a random subset of $[n]$ of size $\tau$ chosen uniformly at random. Then $p_i = \tau/n$ for all $i \in [n]$. Moreover, let $\binom{n}{\tau}$ represents the number of combinations of $n$ taken $\tau$ at a time, $p_C = \frac{1}{\binom{n}{\tau}}$ whenever $|C| = \tau$ and $p_C = 0$ otherwise. So,

$$\mu_{\mathrm{AS}} = \mu_{\mathrm{NICE}}(\tau) := \min_{C \subseteq [n], p_C > 0} \sum_{i \in C} \frac{\mu_i}{n p_i} = \min_{C \subseteq [n], |C| = \tau} \frac{1}{\tau} \sum_{i \in C} \mu_i,$$

$$\sigma_{\star,\mathrm{AS}}^2 = \sigma_{\star,\mathrm{NICE}}^2(\tau) := \sum_{C \subseteq [n], p_C > 0} p_C \left\| \nabla f_C (x_\star) \right\|^2 \stackrel{Eqn.\ (2)}{=} \sum_{C \subseteq [n], |C| = \tau} \frac{1}{\binom{n}{\tau}} \left\| \frac{1}{\tau} \sum_{i \in C} \nabla f_i (x_\star) \right\|^2.$$

It can be shown that $\mu_{\mathrm{NICE}}(\tau)$ is a *nondecreasing* function of $\tau$ (Appendix F.6). So, as the minibatch size $\tau$ increases, the strong convexity constant $\mu_{\mathrm{NICE}}(\tau)$ can only improve. Since $\mu_{\mathrm{NICE}}(1) = \min_i \mu_i$ and $\mu_{\mathrm{NICE}}(n) = \frac{1}{n} \sum_{i=1}^n \mu_i$, the value of $\mu_{\mathrm{NICE}}(\tau)$ interpolates these two extreme cases as $\tau$ varies between 1 and $n$. Conversely, $\sigma_{\star,\mathrm{NICE}}^2(\tau) = \frac{n/\tau - 1}{n-1} \sigma_{\star,\mathrm{NICE}}^2(1)$ is a nonincreasing function, reaching a value of $\sigma_{\star,\mathrm{NICE}}^2(n) = 0$, as explained in Appendix F.6.

**Block Sampling (BS).** Let $C_1, \ldots, C_b$ be a partition of $[n]$ into $b$ nonempty blocks. For each $i \in [n]$, let $B(i)$ indicate which block $i$ belongs to. In other words, $i \in C_j$ if $B(i) = j$. Let $S = C_j$ with probability $q_j > 0$, where $\sum_j q_j = 1$. Then $p_i = q_{B(i)}$, and hence Equation (5) takes on the form

$$\mu_{\mathrm{AS}} = \mu_{\mathrm{BS}} := \min_{j \in [b]} \frac{1}{n q_j} \sum_{i \in C_j} \mu_i, \quad \sigma_{\star,\mathrm{AS}}^2 = \sigma_{\star,\mathrm{BS}}^2 := \sum_{j \in [b]} q_j \left\| \sum_{i \in C_j} \frac{1}{n p_i} \nabla f_i (x_\star) \right\|^2.$$

*Considering two extreme cases:* If $b = 1$, then SPPM-BS = SPPM-FS = PPM. So, indeed, we recover the same rate as SPPM-FS. If $b = n$, then SPPM-BS = SPPM-NS. So, indeed, we recover the same rate as SPPM-NS. We provide the detailed analysis in Appendix B.3.

**Stratified Sampling (SS).** Let $C_1, \ldots, C_b$ be a partition of $[n]$ into $b$ nonempty blocks, as before. For each $i \in [n]$, let $B(i)$ indicate which block does $i$ belong to. In other words, $i \in C_j$ iff $B(i) = j$. Now, for each $j \in [b]$ pick $\xi_j \in C_j$ uniformly at random, and define $S = \cup_{j \in [b]} \{\xi_j\}$. Clearly, $p_i = \frac{1}{|C_{B(i)}|}$. Let's denote $\mathbf{i}_b := (i_1, \cdots, i_b)$, $\mathbf{C}_b := C_1 \times \cdots \times C_b$. Then, Equation (5) take on the form

$$\mu_{\mathrm{AS}} = \mu_{\mathrm{SS}} := \min_{\mathbf{i}_b \in \mathbf{C}_b} \sum_{j=1}^b \frac{\mu_{i_j} |C_j|}{n}, \quad \sigma_{\star,\mathrm{AS}}^2 = \sigma_{\star,\mathrm{SS}}^2 := \sum_{\mathbf{i}_b \in \mathbf{C}_b} \left( \prod_{j=1}^b \frac{1}{|C_j|} \right) \left\| \sum_{j=1}^b \frac{|C_j|}{n} \nabla f_{i_j} (x_\star) \right\|^2.$$

*Considering two extreme cases:* If $b = 1$, then SPPM-SS = SPPM-US. So, indeed, we recover the same rate as SPPM-US. If $b = n$, then SPPM-SS = SPPM-FS. So, indeed, we recover the same rate as SPPM-FS. We provide the detailed analysis in Appendix B.3.

### 2.4 Comparing Stratified Sampling with Block Sampling and Nice Sampling

**Lemma 1** (Stratified Sampling Variance Bounds). Consider the stratified sampling. For each $j \in [b]$, define

$$\sigma_j^2 := \max_{i \in C_j} \left\| \nabla f_i(x_\star) - \frac{1}{|C_j|} \sum_{l \in C_j} \nabla f_l(x_\star) \right\|^2.$$

In words, $\sigma_j^2$ is the maximal squared distance of a gradient (at the optimum) from the mean of the gradients (at optimum) within cluster $C_j$. Then

$$\sigma_{\star,\mathrm{SS}}^2 \leq \frac{b}{n^2} \sum_{j=1}^b |C_j|^2 \sigma_j^2 \leq b \max\left\{ \sigma_1^2, \ldots, \sigma_b^2 \right\}.$$

Note that Lemma 1 provides insights into how the variance might be reduced through stratified sampling. For instance, in a scenario of complete inter-cluster homogeneity, where $\sigma_j^2 = 0$ for all $j$, both bounds imply that $0 = \sigma_{\star,\mathrm{SS}}^2 \leq \sigma_{\star,\mathrm{BS}}^2$. Thus, in this scenario, the convergence neighborhood of stratified sampling is better than that of block sampling.

**Stratified Sampling Outperforms Block Sampling and Nice Sampling in Convergence Neighborhood.** We theoretically compare stratified sampling with block sampling and nice sampling, advocating for stratified sampling as the superior method for future clustering experiments due to its optimal variance properties. We begin with the assumption of $b$ clusters of uniform size $b$ (Assumption F.10), which simplifies the analysis by enabling comparisons of various sampling methods, all with the same sampling size, $b$: $b$-nice sampling, stratified sampling with $b$ clusters, and block sampling where all clusters are of uniform size $b$. Furthermore, we introduce the concept of optimal clustering for stratified sampling (noted as $\mathcal{C}_{b,\mathrm{SS}}$, Definition F.11) in response to a counterexample where block sampling and nice sampling achieve lower variance than stratified sampling (Appendix F.8). Finally, with Assumption F.10 and Definition F.11 in place, we can compare the convergence neighborhoods of stratified sampling and nice sampling.

**Lemma 2.** Given Assumption F.10, the following holds: $\sigma_{\star,\mathrm{SS}}^2(\mathcal{C}_{b,\mathrm{SS}}) \leq \sigma_{\star,\mathrm{NICE}}^2$ for arbitrary $b$. Moreover, the variance within the convergence neighborhood of stratified sampling is less than or equal to that of nice sampling: $\frac{\gamma \sigma_{\star,\mathrm{SS}}^2}{\gamma \mu_{\mathrm{SS}}^2 + 2\mu_{\mathrm{SS}}}(\mathcal{C}_{b,\mathrm{SS}}) \leq \frac{\gamma \sigma_{\star,\mathrm{NICE}}^2}{\gamma \mu_{\mathrm{NICE}}^2 + 2\mu_{\mathrm{NICE}}}$.

Lemma 2 demonstrates that, under specific conditions, the stratified sampling neighborhood is preferable to that of nice sampling. One might assume that, under the same assumptions, a similar assertion could be made for showing that block sampling is inferior to stratified sampling. However, this has only been theoretically verified for the simplified case where both the block size and the number of blocks are $b = 2$, as detailed in Appendix F.8.

## 3 Experiments

**Practical Decision-Making with SPPM-AS.** In our analysis of SPPM-AS, guided by theoretical foundations of Theorem 1 and empirical evidence summarized in Table 1, we explore practical decision-making for varying scenarios. This includes adjustments in hyperparameters within the framework $KT(\epsilon, \mathcal{S}, \gamma, \mathcal{A}(K))$. Here, $\epsilon$ represents accuracy goal, $\mathcal{S}$ represents the sampling distribution, $\gamma$ is representing global learning rate (proximal operator parameter), $\mathcal{A}$ denotes the proximal optimization algorithm, while $K$ denotes the number of local communication rounds. In table 1,

Table 1: $KT(\epsilon, \mathcal{S}, \gamma, \mathcal{A}(K))$

| HP | Control | $KT(\cdots)$ | Exp. |
|---|---|---|---|
| $\gamma$ | $\gamma \uparrow$ | $KT \downarrow, \epsilon \uparrow$ [1] | D.2 |
| | optimal $(\gamma, K) \uparrow$ | $\downarrow$ | 3.3 |
| $\mathcal{A}$ | $\mu$-convex + BFGS/CG | $\downarrow$ compared to LocalGD | 3.3 |
| | NonCVX and Hierarchical FL + Adam with tuned lr | $\downarrow$ compared to LocalGD | 3.7 |

[1] $\epsilon$ is a convergence neighborhood or accuracy.

we summarize how changes on following hyperparameters will influence target metric. With increasing learning rate $\gamma$ one achieves faster convergence with smaller accuracy, also noted as accuracy-rate tradeoff. Our primary observation that with an increase in both the learning rate, $\gamma$, and the number of local steps, $K$, leads to an improvement in the convergence rate. Employing various local solvers for proximal operators also shows an improvement in the convergence rate compared to FedAvg in both convex and non-convex cases.

**Inexact Prox Implementation.**  In practice, the proximal operator cannot be calculated exactly, as proposed in the theoretical version of SPPM-AS (Algorithm 1). In our work, we tackle two approaches for estimating the proximal operator. For logistic regression, we use a simplified approach that employs a virtual hub for computation. When integrated into a hierarchical FL architecture with physical hubs, this approach minimizes communication costs. Standard optimization algorithms such as BFGS and CG are applied to handle the proximal operations. In the neural network experiments, we use local optimization algorithms to estimate the proximal operator. We treat the argument of the proximal operator as an optimization objective and decompose it into functions corresponding to each worker: $x_{t+1} = \text{prox}_{\gamma f_{s_t}}(x_t) := \arg\min_y \left( f_{s_t}(y) + \frac{1}{2\gamma}|y - x|^2 \right) = \arg\min_y \left( \sum_{i \in S_t} f_i(y) + \frac{1}{2\gamma|S_t|}|y - x|^2 \right) = \arg\min_y \left( \sum_{i \in S_t} \tilde{f}_i(y) \right)$ Thus, various local methods for minimizing $\min_y \left( \sum_{i \in S_t} \tilde{f}_i(y) \right)$ can be applied, as detailed in Appendix A.3.

## 3.1 OBJECTIVE AND DATASETS

Our analysis begins with logistic regression with a convex $l_2$ regularizer, which can be represented as:

$$f_i(x) := \frac{1}{n_i} \sum_{j=1}^{n_i} \log\left(1 + \exp(-b_{i,j} x^T a_{i,j})\right) + \frac{\mu}{2}\|x\|^2,$$

where $\mu$ is the regularization parameter, $n_i$ denotes the total number of data points at client $i$, $a_{i,j}$ are the feature vectors, and $b_{i,j} \in \{-1, 1\}$ are the corresponding labels. Each function $f_i$ exhibits $\mu$-strong convexity and $L_i$-smoothness, with $L_i$ computed as $\frac{1}{4n_i} \sum_{j=1}^{n_i} \|a_{i,j}\|^2 + \mu$. For our experiments, we set $\mu$ to 0.1.

Our study utilized datasets from the LibSVM repository (Chang & Lin, 2011), including `mushrooms`, `a6a`, `ijcnn1.bz2`, and `a9a`. We divided these into feature-wise heterogeneous non-iid splits for FL, detailed in Appendix C.1, with a default cohort size of 10. We primarily examined logistic regression, finding results consistent with our theoretical framework, as discussed extensively in Section 3.3 through Appendix D.2. Additional neural network experiments are detailed in Section 3.7 and Appendix E.

## 3.2 ON CHOOSING SAMPLING STRATEGY

As shown in Section 2.3, multiple sampling techniques exist. We propose using clustering approach in conjuction with SPPM-SS as the default sampling strategy for all our experiments. The stratified sampling optimal clustering is impractical due to the difficulty in finding $x_\star$; therefore, we employ a clustering heuristic that aligns with the concept of creating homogeneous worker groups. One such method is K-means, which we use by default. More details on our clustering approach can be found in the Appendix C.1. We compare various sampling techniques in the left panel of Figure 3. Extensive ablations verified the efficiency of stratified sampling over other strategies, due to variance reduction (Lemma 1).

## 3.3 COMMUNICATION COST REDUCTION THROUGH INCREASED LOCAL COMMUNICATION ROUNDS

In this study, we investigate whether increasing the number of local communication rounds, denoted as $K$, in our proposed algorithm SPPM-SS, can lead to a decrease in the total communication cost

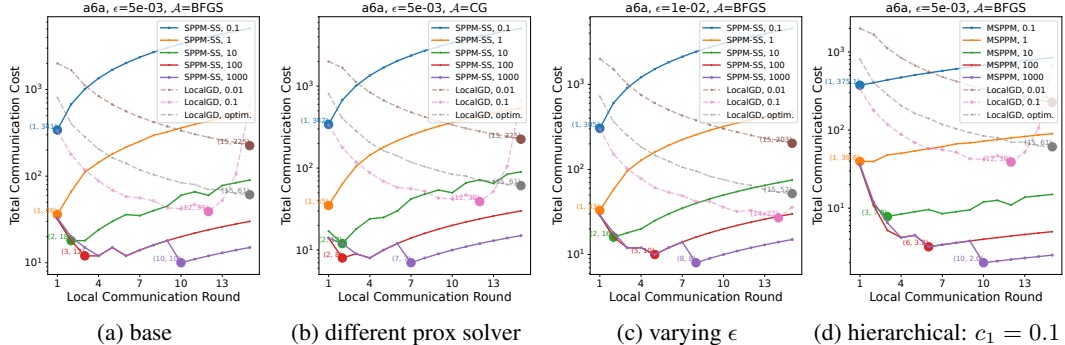

(a) base      (b) different prox solver      (c) varying $\epsilon$      (d) hierarchical: $c_1 = 0.1$

Figure 2: Analysis of total communication costs against local communication rounds for computing the proximal operator. For LocalGD, we align the x-axis to the total local iterations, highlighting the absence of local communication. The aim is to minimize total communication for achieving a predefined global accuracy $\epsilon$, where $\|x_T - x_\star\|^2 < \epsilon$. The optimal step size and minibatch sampling setup for LocalGD are denoted as LocalGD, optim. This showcases a comparison across varying $\epsilon$ values and proximal operator solvers (CG and BFGS).

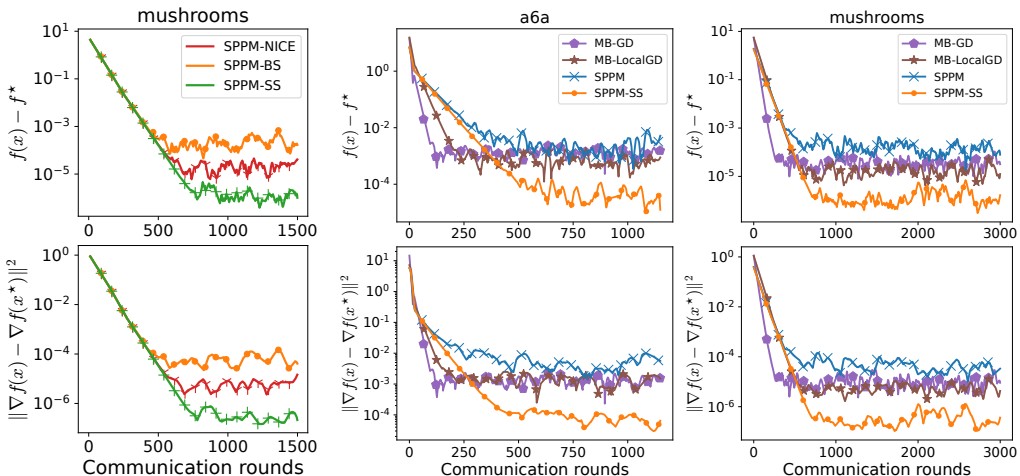

Figure 3: The first column compares sampling methods, while the right two columns analyze convergence relative to popular baselines. $\gamma = 1.0$.

required to converge to a predetermined global accuracy $\epsilon > 0$. In Figure 1, we analyzed various datasets, including a6a and mushrooms, confirming that higher local communication rounds reduce communication costs, especially with larger learning rates. Our study includes both self-ablation of SPPM-SS across different learning rate scales and comparisons with the widely-used cross-device FL method LocalGD (or FedAvg) on the selected cohort. Ablation studies were conducted with a large empirical learning rate of $0.1$, a smaller rate of $0.01$, and an optimal rate as discussed by Khaled & Richtárik (2023), alongside minibatch sampling described by Gower et al. (2019).

In Figure 2, we present more extensive ablations. Specifically, we set the base method (Figure 2a) using the dataset a6a, a proximal solver BFGS, and $\epsilon = 5 \cdot 10^{-3}$. In Figure 2b, we explore the use of an alternative solver, CG (Conjugate Gradient), noting some differences in outcomes. For instance, with a learning rate $\gamma = 1000$, the optimal $K$ with CG becomes 7, lower than 10 in the base setting using BFGS. In Figure 2c, we investigate the impact of varying $\epsilon = 10^{-2}$. Our findings consistently show SPPM-SS's significant performance superiority over LocalGD.

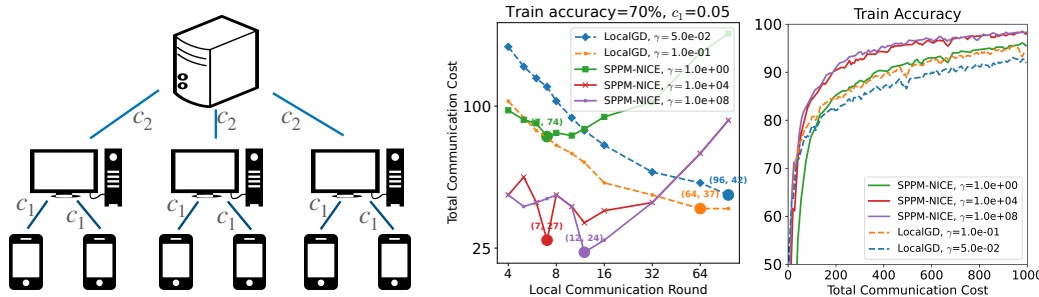

Figure 4: The left column shows the Server-hub-client hierarchical FL architecture. For the right two columns: on the left, communication cost for achieving 70% accuracy in hierarchical FL ($c_1 = 0.05$, $c_2 = 1$); on the right, convergence with optimal hyperparameters ($c_1 = 0.05$, $c_2 = 1$).

### 3.4 EVALUATING THE PERFORMANCE OF VARIOUS SOLVERS $\mathcal{A}$

We further explore the impact of various solvers on optimizing the proximal operators, showcasing representative methods in Table 2 in the Appendix A.3. A detailed overview and comparison of local optimizers listed in the table are provided in Section A.3, given the extensive range of candidate options available. To emphasize key factors, we compare the performance of first-order methods, such as the Conjugate Gradient (CG) method (Hestenes et al., 1952), against second-order methods, like the Broyden-Fletcher-Goldfarb-Shanno (BFGS) algorithm (Broyden, 1967; Shanno, 1970), in the context of strongly convex settings. For non-convex settings, where first-order methods are prevalent in deep learning experiments, we examine an ablation among popular first-order local solvers, specifically choosing MimeLite (Karimireddy et al., 2020a) and FedOpt (Reddi et al., 2020). The comparisons of different solvers for strongly convex settings are presented in Figure 2b, with the non-convex comparison included in the appendix. Upon comparing first-order and second-order solvers in strongly convex settings, we observed that CG outperforms BFGS for our specific problem. In neural network experiments, MimeLite-Adam was found to be more effective than FedOpt variations. However, it is important to note that all these solvers are viable options that have led to impressive performance outcomes.

### 3.5 COMPARATIVE ANALYSIS WITH BASELINE ALGORITHMS

In this section, we conduct an extensive comparison with several established cross-device FL baseline algorithms. Specifically, we examine MB-GD (MiniBatch Gradient Descent with partial client participation), and MB-LocalGD, which is the local gradient descent variant of MB-GD. We default the number of local iterations to 5 and adopt the optimal learning rate as suggested by Gower et al. (2019). To ensure a fair comparison, the cohort size $|C|$ is fixed at 10 for all minibatch methods, including our proposed SPPM-SS. The results of this comparative analysis are depicted in Figure 3. Our findings reveal that SPPM-SS consistently achieves convergence within a significantly smaller neighborhood when compared to the existing baselines. Notably, in contrast to MB-GD and MB-LocalGD, SPPM-SS is capable of utilizing arbitrarily large learning rates. This attribute allows for faster convergence, although it does result in a larger neighborhood size.

### 3.6 HIERARCHICAL FEDERATED LEARNING

We extend our analysis to a hub-based hierarchical FL structure, as conceptualized in the left part of Figure 4. This structure envisions a cluster directly connected to $m$ hubs, with each hub $m_i$ serving $n_i$ clients. The clients, grouped based on criteria such as region, communicate exclusively with their respective regional hub, which in turn communicates with the central server. Given the inherent nature of this hierarchical model, the communication cost $c_1$ from each client to its hub is consistently lower than the cost $c_2$ from each hub to the server. We define communication from clients to hubs as *local communication* and from hubs to the server as *global communication*. Under SPPM-SS, the total cost is expressed as $(c_1 K + c_2)T_{\text{SPPM-SS}}$, while for LocalGD, it is $(c_1 + c_2)T_{\text{LocalGD}}$. As established in Section 3.3, $T_{\text{SPPM-SS}}$ demonstrates significant improvement in total communication costs compared to LocalGD within a hierarchical setting. Our objective is to illustrate this by con-

trasting the standard FL setting, depicted in Figure 2a with parameters $c_1 = 1$ and $c_2 = 0$, against the hierarchical FL structure, which assumes $c_1 = 0.1$ and $c_2 = 1$, as shown in Figure 2d. Given the variation in $c_1$ and $c_2$ values between these settings, a direct comparison of absolute communication costs is impractical. Therefore, our analysis focuses on the ratio of communication cost reduction in comparison to LocalGD. For the `base` setting, LocalGD's optimal total communication cost is 39 with 12 local iterations, whereas for SPPM-SS ($\gamma = 1000$), it is reduced to 10 with 10 local and 1 global communication rounds, amounting to a 74.36% reduction. With the hierarchical FL structure in Figure 2d, SPPM-SS achieves an even more remarkable communication cost reduction of 94.87%. Further ablation studies on varying local communication cost $c_1$ in the Appendix D.3 corroborate these findings.

### 3.7 Neural Network Evaluations

Our empirical analysis includes experiments on Convolutional Neural Networks (CNNs) using the `FEMNIST` dataset, as described by Caldas et al. (2018). We designed the experiments to include a total of 100 clients, with each client representing data from a unique user, thereby introducing natural heterogeneity into our study. We employed the Nice sampling strategy with a cohort size of 10. In contrast to logistic regression models, here we utilize training accuracy as a surrogate for the target accuracy $\epsilon$. For the optimization of the proximal operator, we selected the Adam optimizer, with the learning rate meticulously fine-tuned over a linear grid. Detailed descriptions of the training procedures and the CNN architecture are provided in the Appendix E.

In the deep learning context, we performed a set of experiments similar to those conducted for the convex case. In Appendix E.2, we review nice, block, and stratified sampling strategies, demonstrating the superiority of stratified sampling. Additionally, Appendix E.4 compares various local solvers for the proximal operator. For comparison with the baselines our analysis primarily focuses on the hierarchical FL structure. Initially, we draw a comparison between our proposed method, SPPM-AS, and LocalGD. The crux of our investigation is the total communication cost required to achieve a predetermined level of accuracy, with findings detailed in the right part of Figure 4. Significantly, SPPM-AS demonstrates enhanced performance with the integration of multiple local communication rounds. Notably, the optimal number of these rounds tends to increase alongside the parameter $\gamma$. For each configuration, the convergence patterns corresponding to the sets of optimally tuned hyperparameters are depicted in Figure 4.

## 4 Conclusion

Our research challenges the conventional single-round communication model in federated learning by presenting a novel approach where cohorts participate in multiple communication rounds. This adjustment leads to a significant 74% reduction in communication costs, underscoring the efficacy of extending cohort engagement beyond traditional limits. Our method, SPPM-AS, equipped with diverse client sampling procedures, contributes substantially to this efficiency. This foundational work showcases a pivotal shift in federated learning strategies. Future work could focus on improving algorithmic robustness and ensuring privacy compliance.

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

CONTENTS

# A    RELATED WORK

## A.1    CROSS-DEVICE FEDERATED LEARNING

This paper delves into the realm of Federated Learning (FL), focusing on the cross-device variant, which presents unique and significant challenges. In FL, two predominant settings are recognized: cross-silo and cross-device scenarios, as detailed in Table 1 of Kairouz et al., 2019. The primary distinction lies in the nature of the clients: cross-silo FL typically involves various organizations holding substantial data, whereas cross-device FL engages a vast array of mobile or IoT devices. In cross-device FL, the complexity is heightened by the inability to maintain a persistent hidden state for each client, unlike in cross-silo environments. This factor renders certain approaches impractical, particularly those reliant on stateful clients participating consistently across all rounds. Given the sheer volume of clients in cross-device FL, formulating and analyzing outcomes in an expectation form is more appropriate, but more complex than in finite-sum scenarios.

The pioneering and perhaps most renowned algorithm in cross-device FL is FedAvg (McMahan et al., 2017) and implemented in applications like Google's mobile keyboard (Hard et al., 2018; Yang et al., 2018; Ramaswamy et al., 2019). However, it is noteworthy that popular accelerated training algorithms such as Scaffold (Karimireddy et al., 2020b) and ProxSkip (Mishchenko et al., 2022b) are not aligned with our focus due to their reliance on memorizing the hidden state for each client, which is applicable for cross-device FL. Our research pivots on a novel variant within the cross-device framework. Once the cohort are selected for each global communication round, these cohorts engage in what we term as 'local communications' multiple times. The crux of our study is to investigate whether increasing the number of local communication rounds can effectively reduce the total communication cost to converge to a targeted accuracy.

## A.2    STOCHASTIC PROXIMAL POINT METHOD

Our exploration in this paper centers on the Stochastic Proximal Point Method (SPPM), a method extensively studied for its convergence properties. Initially termed as the incremental proximal point method by Bertsekas (2011), it was shown to converge nonasymptotically under the assumption of Lipschitz continuity for each $f_i$. Following this, Ryu & Boyd (2016) examined the convergence rates of SPPM, noting its resilience to inaccuracies in learning rate settings, contrasting with the behavior of Stochastic Gradient Descent (SGD). Further developments in SPPM's application were seen in the works of Patrascu & Necoara (2018), who analyzed its effectiveness in constrained optimization, incorporating random projections. Asi & Duchi (2019) expanded the scope of SPPM by studying a generalized method, AProx, providing insights into its stability and convergence rates under convex conditions. The research by Asi et al. (2020) and Chadha et al. (2022) further extended these findings, focusing on minibatching and convergence under interpolation in the AProx framework.

In the realm of federated learning, particularly concerning non-convex optimization, SPPM is also known as FedProx, as discussed in works like those of Li et al. (2020a) and Yuan & Li (2022). However, it is noted that in non-convex scenarios, the performance of FedProx/SPPM in terms of convergence rates does not surpass that of SGD. Beyond federated learning, the versatility of SPPM is evident in its application to matrix and tensor completion such as in the work of Bumin & Huang (2021). Moreover, SPPM has been adapted for efficient implementation in a variety of optimization problems, as shown by Shtoff (2022). While non-convex SPPM analysis presents significant challenges, with a full understanding of its convex counterpart still unfolding, recent studies such as the one by Khaled & Jin (2023) have reported enhanced convergence by leveraging second-order similarity. Diverging from this approach, our contribution is the development of an efficient minibatch SPPM method SPPM-AS that shows improved results without depending on such assumptions. Significantly, we also provide the first empirical evidence that increasing local communication rounds in finding the proximal point can lead to a reduction in total communication costs.

## A.3    LOCAL SOLVERS

In the exploration of local solvers for the SPPM-AS algorithm, the focus is on evaluating the performance impact of various inexact proximal solvers within federated learning settings, spanning both strongly convex and non-convex objectives. Here's a simple summary of the algorithms discussed:

| Setting | 1st order | 2nd order |
|---|---|---|
| Strongly-Convex | Conjugate Gradients (CG) Accelerated GD Local GD Scaffnew | BFGS AICN LocalNewton |
| Nonconvex | Mime-Adam FedAdam-AdaGrad FedSpeed | Apollo OASIS |

Table 2: Local optimizers for solving the proximal subproblem.

FedAdagrad-AdaGrad (Wang et al., 2021b): Adapts AdaGrad for both client and server sides within federated learning, introducing local and global corrections to address optimizer state handling and solution bias.

BFGS (Broyden, 1967; Fletcher, 1970; Goldfarb, 1970; Shanno, 1970): A quasi-Newton method that approximates the inverse Hessian matrix to improve optimization efficiency, particularly effective in strongly convex settings but with limitations in distributed implementations.

AICN (Hanzely et al., 2022): Offers a global $O(1/k^2)$ convergence rate under a semi-strong self-concordance assumption, streamlining Newton's method without the need for line searches.

LocalNewton (Bischoff et al., 2023): Enhances local optimization steps with second-order information and global line search, showing efficacy in heterogeneous data scenarios despite a lack of extensive theoretical grounding.

Fed-LAMB (Karimi et al., 2022): Extends the LAMB optimizer to federated settings, incorporating layer-wise and dimension-wise adaptivity to accelerate deep neural network training.

FedSpeed (Sun et al., 2023): Aims to overcome non-vanishing biases and client-drift in federated learning through prox-correction and gradient perturbation steps, demonstrating effectiveness in image classification tasks.

Mime-Adam (Karimireddy et al., 2020a): Mitigates client drift in federated learning by integrating global optimizer states and an SVRG-style correction term, enhancing the adaptability of Adam to distributed settings.

OASIS (Jahani et al., 2021): Utilizes local curvature information for gradient scaling, providing an adaptive, hyperparameter-light approach that excels in handling ill-conditioned problems.

Apollo (Ma, 2020): A quasi-Newton method that dynamically incorporates curvature information, showing improved efficiency and performance over first-order methods in deep learning applications.

Each algorithm contributes uniquely to the landscape of local solvers in federated learning, ranging from enhanced adaptivity and efficiency to addressing specific challenges such as bias, drift, and computational overhead.

# B  THEORETICAL OVERVIEW AND RECOMMENDATIONS

## B.1  PARAMETER CONTROL

We have explored the effects of changing the hyperparameters of SPPM-AS on its theoretical properties, as summarized in Table 3. This summary shows that as the learning rate increases, the number of iterations required to achieve a target accuracy decreases, though this comes with an increase in neighborhood size. Focusing on sampling strategies, for SPPM-NICE employing NICE sampling, an increase in the sampling size $\tau_{\mathcal{S}}$ results in fewer iterations ($T$) and a smaller neighborhood. Furthermore, given that stratified sampling outperforms both block sampling and NICE sampling, we recommend adopting stratified sampling, as advised by Lemma 1.

Table 3: Theoretical summary

| Hyperparameter | Control | Rate (T) | Neighborhood |
|---|---|---|---|
| $\gamma$ | $\uparrow$ | $\downarrow$ | $\uparrow$ |
| $\mathcal{S}$ | $\tau_{\mathcal{S}} \uparrow$ [(1)] | $\downarrow$ | $\downarrow$ |
| | Stratified sampling optimal clustering instead of BS or NICE sampling | $\downarrow$ | Lemma 1 |

[(1)] We define $\tau_{\mathcal{S}} := \mathbb{E}_{S \sim \mathcal{S}}\left[|S|\right]$.

Table 4: Arbitrary samplings comparison.

| Setting/Requirement | $\mu_{\mathrm{AS}}$ | $\sigma_{\star,\mathrm{AS}}$ |
|---|---|---|
| Full | $\frac{1}{n}\sum_{i=1}^{n}\mu_i$ | $0$ |
| Non-Uniform | $\min_i \frac{\mu_i}{np_i}$ | $\frac{1}{n}\sum_{i=1}^{n}\frac{1}{np_i}\|\nabla f_i(x_\star)\|^2$ |
| Nice | $\min_{C\subseteq[n],\|C\|=\tau}\frac{1}{\tau}\sum_{i\in C}\mu_i$ | $\sum_{C\subseteq[n],\|C\|=\tau}\frac{1}{\binom{n}{\tau}}\left\|\frac{1}{\tau}\sum_{i\in C}\nabla f_i(x_\star)\right\|^2$ |
| Block | $\min_{j\in[b]}\frac{1}{nq_j}\sum_{i\in C_j}\mu_i$ | $\sum_{j\in[b]}q_j\left\|\sum_{i\in C_j}\frac{1}{np_i}\nabla f_i(x_\star)\right\|^2$ |
| Stratified | $\min_{\mathbf{i}_b\in\mathbf{C}_b}\sum_{j=1}^{b}\frac{\mu_{i_j}\|C_j\|}{n}$ | $\sum_{\mathbf{i}_b\in\mathbf{C}_b}\left(\prod_{j=1}^{b}\frac{1}{\|C_j\|}\right)\left\|\sum_{j=1}^{b}\frac{\|C_j\|}{n}\nabla f_{i_j}(x_\star)\right\|^2$ |
| | | Upper bound: $\frac{b}{n^2}\sum_{j=1}^{b}\|C_j\|^2\sigma_j^2$ |

## B.2 COMPARISON OF SAMPLING STRATEGIES

**Full Sampling (FS).** Let $S = [n]$ with probability 1. Then SPPM-AS applied to Equation (9) becomes PPM (Moreau, 1965; Martinet, 1970) for minimizing $f$. Moreover, in this case, we have $p_i = 1$ for all $i \in [n]$ and Equation (5) takes on the form

$$\mu_{\mathrm{AS}} = \mu_{\mathrm{FS}} := \frac{1}{n}\sum_{i=1}^{n}\mu_i, \quad \sigma_{\star,\mathrm{AS}}^2 = \sigma_{\star,\mathrm{FS}}^2 := 0.$$

Note that $\mu_{\mathrm{FS}}$ is the strong convexity constant of $f$, and that the neighborhood size is zero, as we would expect.

**Nonuniform Sampling (NS).** Let $S = \{i\}$ with probability $p_i > 0$, where $\sum_i p_i = 1$. Then Equation (5) takes on the form

$$\mu_{\mathrm{AS}} = \mu_{\mathrm{NS}} := \min_i \frac{\mu_i}{np_i}, \quad \sigma_{\star,\mathrm{AS}}^2 = \sigma_{\star,\mathrm{NS}}^2 := \frac{1}{n}\sum_{i=1}^{n}\frac{1}{np_i}\|\nabla f_i(x_\star)\|^2.$$

If we take $p_i = \frac{\mu_i}{\sum_{j=1}^{n}\mu_j}$ for all $i \in [n]$, we shall refer to Algorithm 1 as SPPM with importance sampling (SPPM-IS). In this case,

$$\mu_{\mathrm{NS}} = \mu_{\mathrm{IS}} := \frac{1}{n}\sum_{i=1}^{n}\mu_i, \quad \sigma_{\star,\mathrm{NS}}^2 = \sigma_{\star,\mathrm{IS}}^2 := \frac{\sum_{i=1}^{n}\mu_i}{n}\sum_{i=1}^{n}\frac{\|\nabla f_i(x_\star)\|^2}{n\mu_i}.$$

This choice maximizes the value of $\mu_{\mathrm{NS}}$ (and hence minimizes the first part of the convergence rate) over the choice of the probabilities.

Table 4 summarizes the parameters associated with various sampling strategies, serving as a concise overview of the methodologies discussed in the main text. This summary facilitates a quick comparison and reference.

## B.3 EXTREME CASES OF BLOCK SAMPLING AND STRATIFIED SAMPLING

**Extreme cases of block sampling.** We now consider two extreme cases:

- If $b = 1$, then SPPM-BS = SPPM-FS = PPM. Let's see, as a sanity check, whether we recover the right rate as well. We have $q_1 = 1, C_1 = [n], p_i = 1$ for all $i \in [n]$, and the expressions for $\mu_{AS}$ and $\sigma^2_{\star, BS}$ simplify to

$$\mu_{BS} = \mu_{FS} := \frac{1}{n} \sum_{i=1}^{n} \mu_i, \sigma^2_{\star, BS} = \sigma^2_{\star, FS} := 0.$$

So, indeed, we recover the same rate as SPPM-FS.

- If $b = n$, then SPPM-BS = SPPM-NS. Let's see, as a sanity check, whether we recover the right rate as well. We have $C_i = \{i\}$ and $q_i = p_i$ for all $i \in [n]$, and the expressions for $\mu_{AS}$ and $\sigma^2_{\star, BS}$ simplify to

$$\mu_{BS} = \mu_{NS} := \min_{i \in [n]} \frac{\mu_i}{n p_i}, \quad \sigma^2_{\star, BS} = \sigma^2_{\star, NS} := \frac{1}{n} \sum_{i=1}^{n} \frac{1}{n p_i} \left\| \nabla f_i (x_\star) \right\|^2.$$

So, indeed, we recover the same rate as SPPM-NS.

**Extreme cases of stratified sampling.** We now consider two extreme cases:

- If $b = 1$, then SPPM-SS = SPPM-US. Let's see, as a sanity check, whether we recover the right rate as well. We have $C_1 = [n], |C_1| = n, \left( \prod_{j=1}^{b} \frac{1}{|C_j|} \right) = \frac{1}{n}$ and hence

$$\mu_{SS} = \mu_{US} := \min_i \mu_i, \quad \sigma^2_{\star, SS} = \sigma^2_{\star, US} := \frac{1}{n} \sum_{i=1}^{n} \left\| \nabla f_i (x_\star) \right\|^2.$$

So, indeed, we recover the same rate as SPPM-US.

- If $b = n$, then SPPM-SS = SPPM-FS. Let's see, as a sanity check, whether we recover the right rate as well. We have $C_i = \{i\}$ for all $i \in [n], \left( \prod_{j=1}^{b} \frac{1}{|C_j|} \right) = 1$, and hence

$$\mu_{SS} = \mu_{FS} := \frac{1}{n} \sum_{i=1}^{n} \mu_i, \quad \sigma^2_{\star, SS} = \sigma^2_{\star, FS} := 0.$$

So, indeed, we recover the same rate as SPPM-FS.

B.4 FEDERATED AVERAGING SPPM BASELINES

In this section we propose two new algorithms based on federated averaging principle. Since to the best of our knowledge there are no federated averaging analyses within the same assumptions, we provide analysis of modified versions of SPPM-AS.

**Averaging on** $\text{prox}_{\gamma f_i}$**.** We introduce FedProx-SPPM-AS (see Algorithm 2), which is inspired by the principles of FedProx (Li et al., 2020a). Unlike the SPPM-AS approach where a proximal operator is computed for the chosen cohort as a whole, in FedProx-SPPM-AS, we compute and then average the proximal operators calculated for each member within the cohort. One can see, that the FedProx is the simple case of this algorithm, when number of local communication rounds $K = 1$.

**Algorithm 2** Proximal Averaging SPPM-AS (FedProx-SPPM-AS)

1: **Input:** starting point $x_{0,0} \in \mathbb{R}^d$, arbitrary sampling distribution $\mathcal{S}$, learning rate $\gamma > 0$, local communication rounds $K$.
2: **for** $t = 0, 1, 2, \cdots, T - 1$ **do**
3:     Sample $S_t \sim \mathcal{S}$
4:     **for** $k = 0, 1, 2, \cdots K - 1$ **do**
5:         $x_{k+1,t} = \sum_{i \in S_t} \frac{1}{|S_t|} \text{prox}_{\gamma f_i}(x_{k,t})$
6:     **end for**
7:     $x_{0,t+1} \leftarrow x_{K,t}$
8: **end for**
9: **Output:** $x_{0,T}$

**Algorithm 3** Federated Averaging SPPM-AS (FedAvg-SPPM-AS)

1: **Input:** starting point $x_{0,0} \in \mathbb{R}^d$, arbitrary sampling distribution $\mathcal{S}$, global learning rate $\gamma > 0$, local learning rate $\alpha > 0$, local communication rounds $K$
2: **for** $t = 0, 1, 2, \cdots, T - 1$ **do**
3:     Sample $S_t \sim \mathcal{S}$
4:     $\forall i \in S_t \ \tilde{f}_{i,t}(x) \leftarrow f_i(x) + \frac{1}{2\gamma} \|x - x_t\|^2$
5:     **for** $k = 0, 1, 2, \cdots K - 1$ **do**
6:         $x_{k+1,t} = \sum_{i \in S_t} \frac{1}{|S_t|} \text{prox}_{\alpha \tilde{f}_{i,t}}(x_{k,t})$
7:     **end for**
8:     $x_{0,t+1} \leftarrow x_{K,t}$
9: **end for**
10: **Output:** $x_{0,T}$

Here, we employ a proof technique similar to that of Theorem 1 and obtain the following convergence.

**Theorem 2** (FedProx convergence). Let the number of local iterations $K = 1$, and assume that Assumption 2.1 (differentiability) and Assumption 2.2 (strong convexity) hold. Let $x_0 \in \mathbb{R}^d$ be an arbitrary starting point. Then, for any $t \geq 0$ and any $\gamma > 0$, the iterates of FedProx-SPPM (as described in Algorithm 2) satisfy:

$$\mathbb{E}\left[\|x_t - x_\star\|^2\right] \leq A_{\mathcal{S}}^t \|x_0 - x_\star\|^2 + \frac{B_{\mathcal{S}}}{1 - A_{\mathcal{S}}},$$

where $A_{\mathcal{S}} \coloneqq \mathbb{E}_{S_t \sim \mathcal{S}}\left[\frac{1}{|S_t|}\sum_{i \in S_t}\frac{1}{1+\gamma\mu_i}\right]$ and $B_{\mathcal{S}} \coloneqq \mathbb{E}_{S_t \sim \mathcal{S}}\left[\frac{1}{|S_t|}\sum_{i \in S_t}\frac{\gamma}{(1+\gamma\mu_i)\mu_i}\|\nabla f_i(x_\star)\|^2\right]$.

Compared to the theoretical analysis for convex functions provided in Li et al. (2020a), our theoretical bound does not rely on the "gradient boundedness" assumption or the $L$-smooth constant.

**Federated averaging for** prox **approximation.** An alternative method involves estimating the proximal operator by averaging the proximal operators calculated for each worker's function. We call it *Federated Averaging Stochastic Proximal Point Method* (FedAvg-SPPM-AS, see Algorithm 3). (FedAvg-SPPM-AS, see Algorithm 3).

After selecting and fixing a sample of workers $S_k$, the main objective is to calculate the proximal operator. This can be accomplished by approximating the proximal calculation with the goal of minimizing $\tilde{f}_S(x) = f_S(x) + \frac{2}{\gamma}\|x - x_t\|^2$. Essentially, this method utilizes FedProx as a local solver for computing the proximal operator. It can be observed that this approach is equivalent to FedProx-SPPM-AS, as at each local step we calculate

$$\text{prox}_{\alpha \tilde{f}_i}(x_{k,t}) \coloneqq \underset{z \in \mathbb{R}^d}{\arg\min}\left[\tilde{f}_i(z) + \frac{2}{\alpha}\|z - x_{k,t}\|^2\right] = \underset{z \in \mathbb{R}^d}{\arg\min}\left[f_i(z) + \left(\frac{2}{\gamma} + \frac{2}{\alpha}\right)\|z - x_{k,t}\|^2\right].$$

It follows that FedProx is equivalent to FedAvg-SPPM-AS when the number of communication rounds is set to $K = 1$. Thus, we can conclude that FedProx is a specific instance of SPPM-AS.

## C    TRAINING DETAILS

### C.1    NON-IID DATA GENERATION

In our study, we validate performance and compare the benefits of SPPM-AS over SPPM using well-known datasets such as `mushrooms`, `a6a`, `w6a`, and `ijcnn1.bz2` from LibSVM (Chang & Lin, 2011). To ensure relevance to our research focus, we adopt a feature-wise non-IID setting, characterized by variation in feature distribution across clients. This variation is introduced by

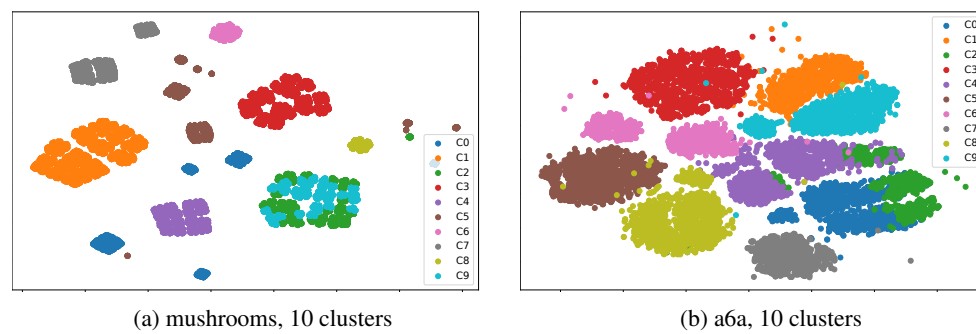

(a) mushrooms, 10 clusters          (b) a6a, 10 clusters

Figure 5: t-SNE visualization of cluster-features across data samples on clients.

clustering the features using the K-means algorithm, with the number of clusters set to 10 and the number of clients per cluster fixed at 10 for simplicity. We visualize the clustered data using t-SNE in Figure 5, where we observe that the data are divided into 10 distinct clusters with significantly spaced cluster centers.

## C.2 SAMPLING

To simulate random sampling among clients within these 10 clusters, where each cluster comprises 10 clients, we consider two contrasting scenarios:

- *Case I - SPPM-BS:* Assuming clients within the same cluster share similar features and data distributions, sampling all clients from one cluster (i.e., $C = 10$ clients) results in a homogeneous sample.
- *Case II - SPPM-SS:* Conversely, by traversing all 10 clusters and randomly sampling one client from each, we obtain a group of 10 clients representing maximum heterogeneity.

We hypothesize that any random sampling from the 100 clients will yield performance metrics lying between these two scenarios. In Figure 6, we examine the impact of sampling clients with varying degrees of heterogeneity using a fixed learning rate of 0.1. Our findings indicate that heterogeneous sampling results in a significantly smaller convergence neighborhood $\sigma_\star^2$. This outcome is attributed to the broader global information captured through heterogeneous sampling, in contrast to homogeneous sampling, which increases the data volume without contributing additional global insights. As these two sampling strategies represent the extremes of arbitrary sampling, any random selection will fall between them in terms of performance. Given their equal cost and the superior performance of the SPPM-SS strategy in heterogeneous FL environments, we designate SPPM-SS as our default sampling approach.

## C.3 SPPM-AS ALGORITHM ADAPTATION FOR FEDERATED LEARNING

In the main text, Algorithm 1 outlines the general form of SPPM-AS. For the convenience of implementation in FL contexts and to facilitate a better understanding, we introduce a tailored version of the SPPM-AS algorithm specific to FL, designated as Algorithm 4. Notably, as stratified sampling is adopted as our default method, this adaptation of the algorithm specifically addresses the nuances of the block sampling approach. We also conducted arbitrary sampling on synthetic datasets and neural networks to demonstrate the algorithm's versatility.

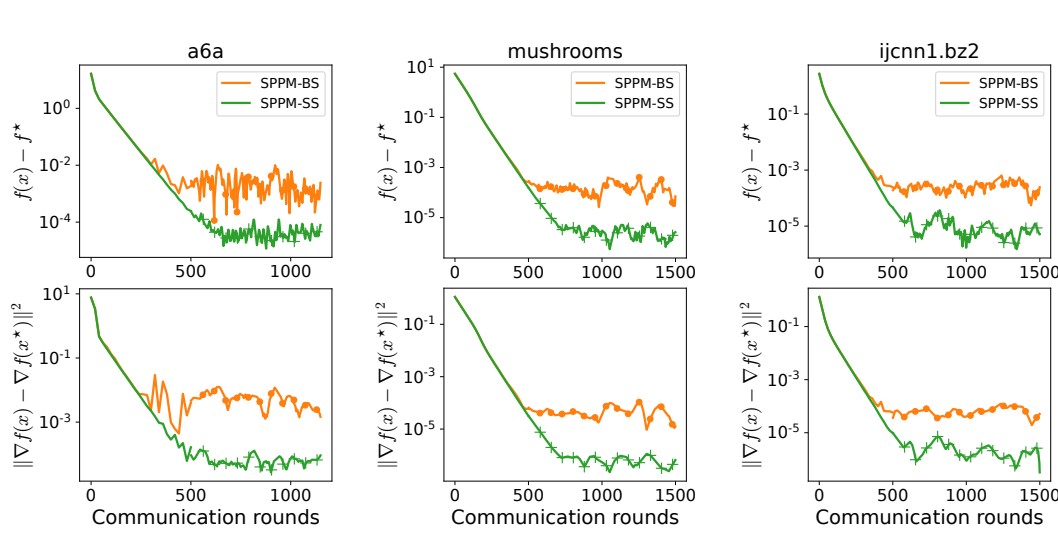

Figure 6: Comparison with SPPM-SS and SPPM-BS samplings.

---

**Algorithm 4** SPPM-AS Adaptation for Federated Learning

---

1: **Input:** Initial point $x^0 \in \mathbb{R}^d$, cohort size $C \geq 1$, learning rate $\gamma > 0$, clusters $q \geq C$, local communication rounds $K$
2: **for** $t = 0, 1, 2, \cdots$ **do**
3:    SPPM-BS:
4:       Server samples a cluster $q_i$ from $[q]$
5:       Server samples $C$ clients, denoted as $[C]$ from cluster $q_i$
6:    SPPM-SS:
7:       Server samples $C$ clusters from $[q]$
8:       Server sample 1 client from each selected cluster to construct $C$ clients
9:    Server broadcasts the model $x_t$ to each $C_i \in [C]$
10:   All selected clients in parallel construct $F_{\xi_t^1, \cdots, \xi_t^C}(x_t)$
11:   All selected clients together evaluate the prox for $K$ local communication rounds to obtain
12:

$$x_{t+1} \simeq \text{prox}_{\gamma F_{\xi_t^1, \cdots, \xi_t^C}}(x_t)$$

13:   All selected clients send the updated model $x_{t+1}$ to the server
14: **end for**

---

# D ADDITIONAL EXPERIMENTS ON LOGISTIC REGRESSION

## D.1 COMMUNICATION COST ON VARIOUS DATASETS TO A TARGET ACCURACY

In Figure 1, we presented the total communication cost relative to the number of rounds required to achieve the target accuracy for the selected cohort. In this section, we provide more details on how is this figure was obtained and present additional results for various datasets.

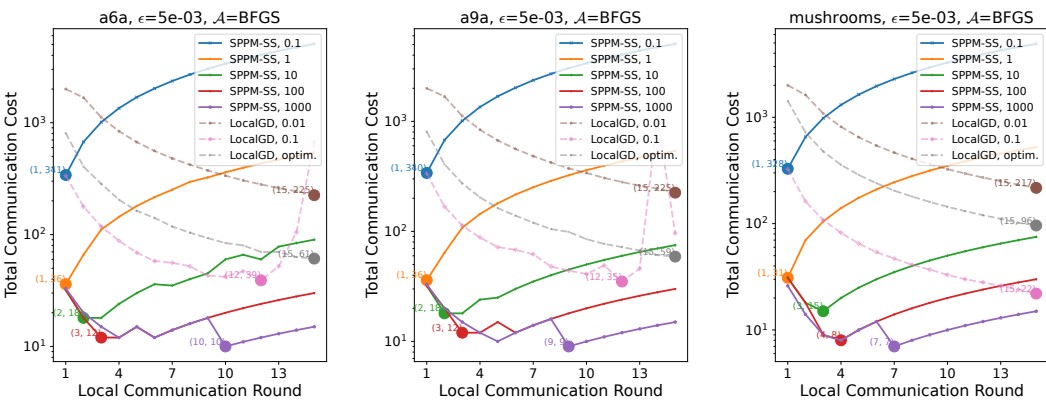

Figure 7: Total communication cost with respect to the local communication round. For LocalGD, $K$ represents the local communication round $K$ for finding the prox of the current model. For LocalGD, we slightly abuse the x-axis, which represents the total number of local iterations, no local communication is required. We calculate the total communication cost to reach a fixed global accuracy $\epsilon$ such that $\|x_t - x_\star\|^2 < \epsilon$. LocalGD, optim represents using the theoretical optimal stepsize of LocalGD with minibatch sampling.

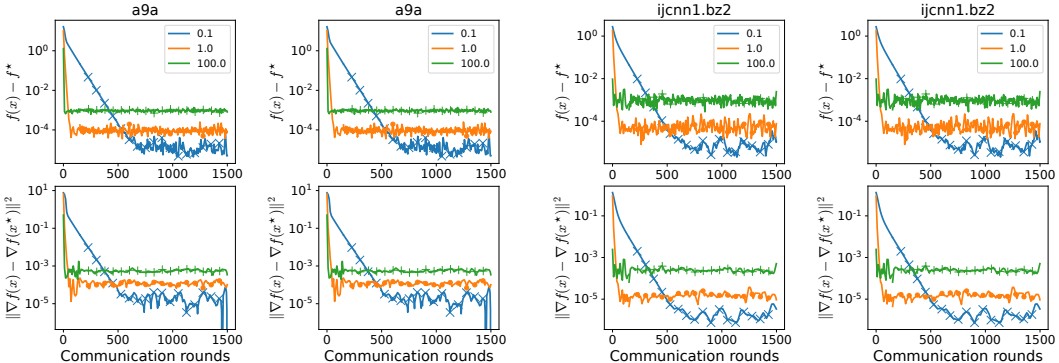

Figure 8: $K = 4$.           Figure 9: $K = 16$.

## D.2 CONVERGENCE SPEED AND $\sigma^2_{\star,\mathrm{SS}}$ TRADE-OFF

Unlike SGD-type methods such as MB-GD and MB-LocalGD, in which the largest allowed learning rate is $1/A$, where $A$ is a constant proportion to the smoothness of the function we want to optimize (Gower et al., 2019). For larger learning rate, SGD-type method may not converge and exploding. However, for stochastic proximal point methods, they have a very descent benefit of allowing arbitrary learning rate. In this section, we verify whether our proposed method can allow arbitrary learning rate and whether we can find something interesting. We considered different learning rate scale from 1e-5 to 1e+5. We randomly selected three learning rates [0.1, 1, 100] for visual representation with the results presented in Figure 8 and Figure 9. We found that a larger learning rate leads to a faster convergence rate but results in a much larger neighborhood, $\sigma^2_{\star,\mathrm{SS}}/\mu^2_{\mathrm{SS}}$. This can be considered a trade-off between convergence speed and neighborhood size, $\sigma^2_{\star,\mathrm{SS}}$. By default,

we consider setting the learning rate to $1.0$ which has a good balance between the convergence speed and the neighborhood size.

In this section, we extend our analysis by providing additional results across a broader range of datasets and varying learning rates. Specifically, Figure 8 illustrates the outcomes using 4 local communication rounds ($K = 4$), while Figure 9 details the results for 16 local communication rounds ($K = 16$). Previously, in Figure 1, we explored the advantages of larger $K$ values. Here, our focus shifts to determining if similar trends are observable across different $K$ values. Through comprehensive evaluations on various datasets and multiple $K$ settings, we have confirmed that lower learning rates in SPPM-AS result in slower convergence speeds; however, they also lead to a smaller final convergence neighborhood.

### D.3 ADDITIONAL EXPERIMENTS ON HIERARCHICAL FEDERATED LEARNING

In Figure 2d of the main text, we detail the total communication cost for hierarchical Federated Learning (FL) utilizing parameters $c_1 = 0.1$ and $c_2 = 1$ on the a6a dataset. Our findings reveal that SPPM-AS achieves a significant reduction in communication costs, amounting to $94.87\%$, compared with the conventional FL setting where $c_1 = 1$ and $c_2 = 1$, which shows a $74.36\%$ reduction. In this section, we extend our analysis with comprehensive evaluations on additional datasets, namely ijcnn1.bz2, a9a, and mushrooms. Beyond considering $c_1 = 0.1$, we further explore the impact of reducing the local communication cost from each client to the corresponding hub to $c_1 = 0.05$. The results, presented in Figure 10 and the continued Figure 11, reinforce our observation: hierarchical FL consistently leads to further reductions in communication costs. A lower $c_1$ parameter correlates with even greater savings in communication overhead. These results not only align with our expectations but also underscore the efficacy of our proposed SPPM-AS in cross-device FL settings.

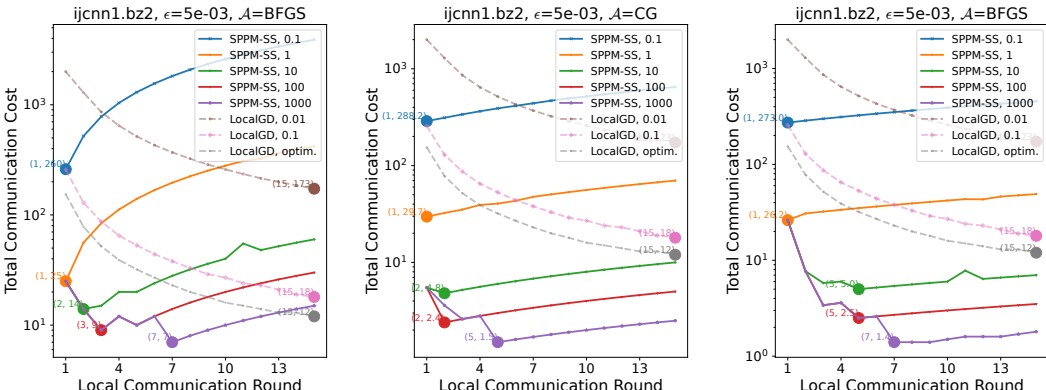

Figure 10: The total communication cost is analyzed with respect to the number of local communication rounds. For LocalGD, $K$ represents the local communication round used for finding the prox of the current model. In the case of LocalGD, we slightly abuse the x-axis to represent the total number of local iterations, as no local communication is required. We calculate the total communication cost needed to reach a fixed global accuracy $\epsilon$, such that $\|x_t - x_\star\|^2 < \epsilon$. LocalGD, optim denotes the use of the theoretically optimal stepsize for LocalGD with minibatch sampling. Comparisons are made between different prox solvers (CG and BFGS).

## E ADDITIONAL NEURAL NETWORK EXPERIMENTS

### E.1 EXPERIMENT DETAILS

For our neural network experiments, we used the FEMNIST dataset (Caldas et al., 2018). Each client was created by uniformly selecting from user from original dataset, inherently introducing heterogeneity among clients. We tracked and reported key evaluation metrics—training and testing loss and accuracy—after every 5 global communication rounds. The test dataset was prepared by

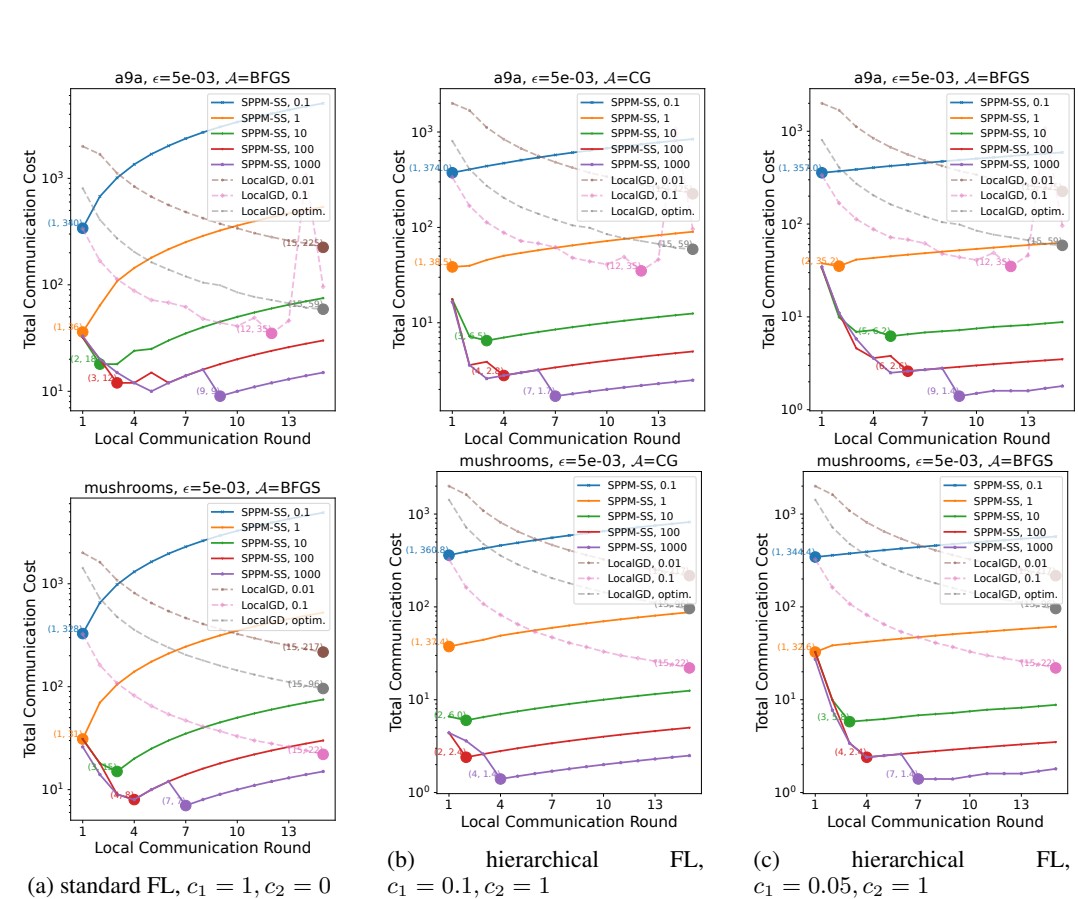

Figure 11: Total communication cost with respect to the local communication round.

| Layer | Output Shape | # of Trainable Parameters | Activation | Hyperparameters |
|---|---|---|---|---|
| Input | (28, 28, 1) | 0 | | |
| Conv2d | (24, 24, 32) | 832 | ReLU | kernel size = 5; strides = (1, 1) |
| Conv2d | (10, 10, 64) | 51,264 | ReLU | kernel size = 5; strides = (1, 1) |
| MaxPool2d | (5, 5, 64) | 0 | | pool size = (2, 2) |
| Flatten | 6400 | 0 | | |
| Dense | 128 | 819,328 | ReLU | |
| Dense | 62 | 7,998 | softmax | |

Table 5: Architecture of the CNN model for FEMNIST symbol recognition.

dividing each user's data into a 9:1 ratio, following the partitioning approach of the FedLab framework (Zeng et al., 2023). For the SPPM-AS algorithm, we selected Adam as the optimizer for the proximal operator. The learning rate was determined through a grid search across the following range: $[0.0001, 0.0005, 0.001, 0.005, 0.01, 0.05, 0.1, 0.5]$. The model architecture comprises a convolutional neural network (CNN) with the following layers: Conv2d(1, 32, 5), ReLU, Conv2d(32, 64, 5), MaxPool2d(2, 2), a fully connected (FC) layer with 128 units, ReLU, and another FC layer with 128 units, as specified in Table 5. Dropout, learning rate scheduling, gradient clipping, etc., were not used to improve the interpretability of results.

We explore various values of targeted training accuracy, as illustrated in Figure 12. This analysis helps us understand the impact of different accuracy thresholds on the model's performance. For instance, we observe that as the target accuracy changes, SPPM-NICE consistently outperforms LocalGD in terms of total communication cost. As the target accuracy increases, the performance gap between these two algorithms also widens. Additionally, we perform ablation studies on different values of $c_1$, as shown in Figure 13, to assess their effects on the learning process. Here, we note that with $c_2 = 0.2$, SPPM-NICE performs similarly to LocalGD, suggesting that an increase in $c_2$ value could narrow the performance gap between SPPM-NICE and LocalGD.

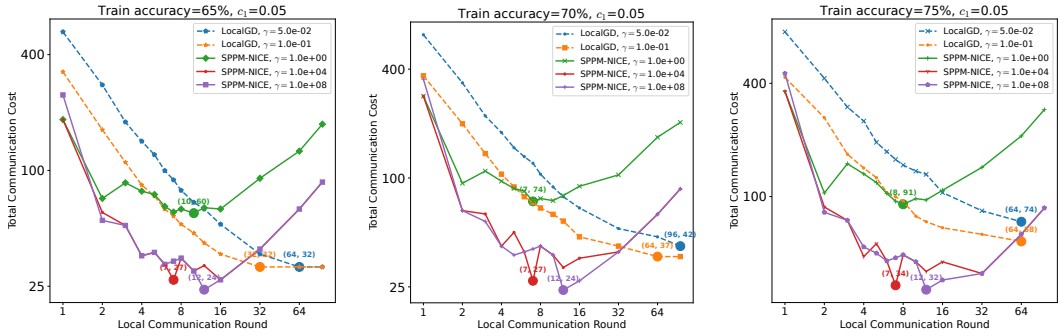

Figure 12: Varying targeted training accuracy level for SPPM-AS.

### E.2 SAMPLINGS DEEP LEARNING PRACTICAL COMPARISON

In the arbitrary samplings theory section 2.3, we referred to several sampling techniques such as batch sampling and nice sampling, which require dividing workers into clusters. We have reviewed the theoretical properties of clustering using the values of gradients at the optimum $x_*$. However, in practice, this information is unavailable, necessitating the use of various clustering methods based on heuristics. To achieve this, we can apply unsupervised algorithms such as feature-wise K-means. In the federated learning scenario, each worker's dataset consists of numerous data points, so we need to create some representation of them. One of the most straightforward methods is to average the dataset points. The 2-dimensional t-SNE representation of the provided clustering is shown in Figure 14. Cluster patterns are visible, but they are not as well separated as theory suggests.

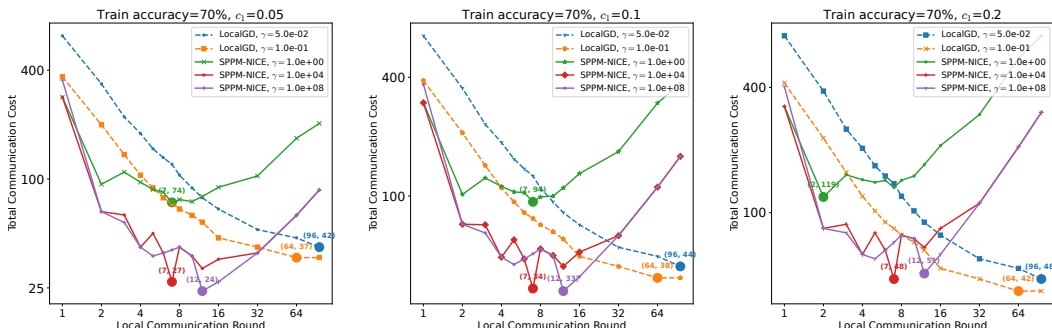

Figure 13: Varying $c_1$ cost.

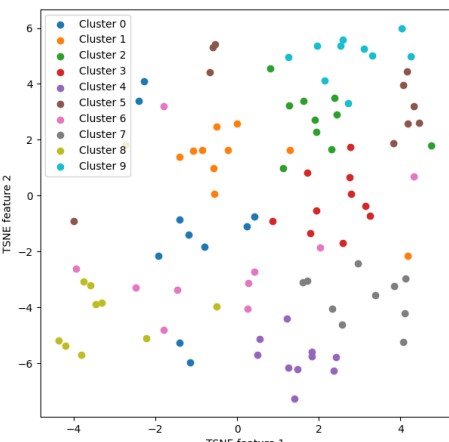

Figure 14: 2-dimensional t-SNE representation of feature-wise averaging K-means clustering of 100 randomly chosen workers from the `FEMNIST` dataset into 10 equally sized clusters.

After applying the clustering, we can compare the performance of the proposed sampling techniques, as shown in Figure 15. The performance gap between the nice and stratified samplings is insignificant, suggesting that clustering is not effectively separating the workers based on feature averaging. However, it is noticeable that stratified sampling consistently outperforms block sampling, demonstrating greater stability. This observation aligns with the hypothesis stated in Appendix F.8.

### E.3 CONVERGENCE ANALYSIS COMPARED WITH BASELINES

Further, we compare SPPM-AS, SPPM, and LocalGD in Figure 17, placing a particular emphasis on evaluating the total computational complexity. This measure gains importance in scenarios where communication rounds are of secondary concern, thereby shifting the focus to the assessment of computational resource expenditure.

### E.4 PROX SOLVERS BASELINES

We compare baselines from A.3 for training a CNN model over 100 workers using data from the `FEMNIST` dataset, as shown in Figure 16. The number of local communication rounds and worker optimizer steps is consistent among various solvers for the purpose of fair comparison. All local solvers optimize the local objective, which is prox on the selected cohort. The solvers compared are: LocalGD referred as FedSGD (McMahan et al., 2017) - the Federated Averaging algorithm with SGD as the worker optimizer, FedAdam - the Federated Averaging algorithm with Adam as the worker optimizer, FedAdam-Adam based on the FedOpt framework (Reddi et al., 2020), and finally

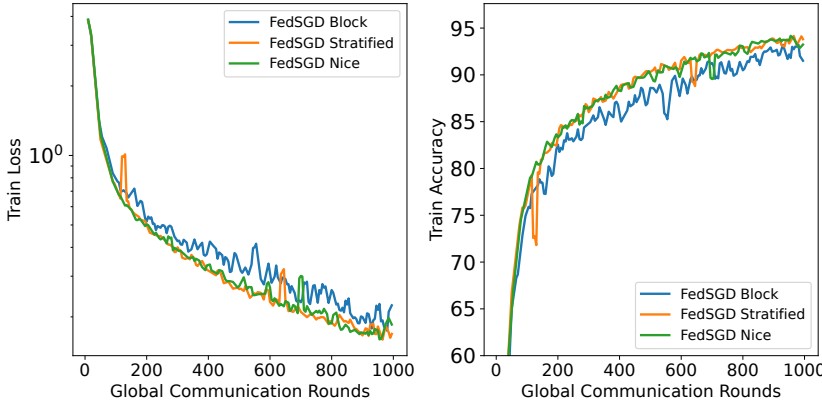

Figure 15: Comparison of stratified, block, and nice samplings based on training data convergence. For stratified and block sampling, feature-wise averaged K-Means clustering into 10 clusters is used. All parameters, aside from samplings, are fixed at the same values: the number of local communication rounds ($T$) is 3, the number of worker training epochs is 3, and the number of sampled workers is 10.

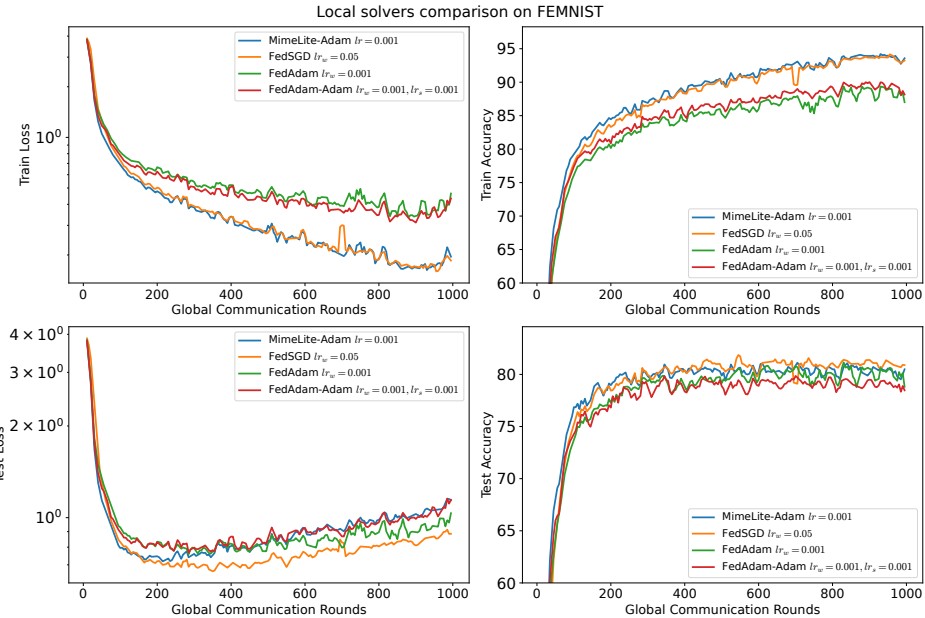

Figure 16: Different local solvers for prox baselines for training a CNN model over 100 workers using data from the FEMNIST dataset. The number of local communication rounds is fixed at 3 and the number of worker optimizer steps is fixed at 3. Nice sampling with a minibatch size of 10 is used. $\gamma$ is fixed at 1.0.

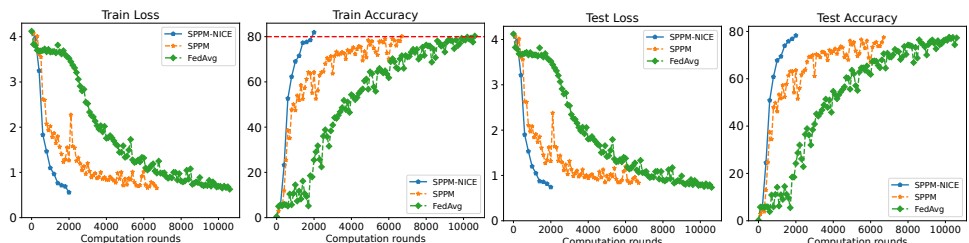

Figure 17: Accuracy compared with baselines.

MimeLite-Adam, which is based on the Mime (Karimireddy et al., 2020a) framework and the Adam optimizer. The hyperparameter search included a double-level sweep of the optimizer learning rates: $[0.00001, 0.0001, 0.001, 0.01, 0.1]$, followed by $[0.25, 0.5, 1.0, 2.5, 5] * lr_{\text{best}}$. One can see that all methods perform similarly, with MimeLite-Adam and FedSGD converging better on the test data.

# F   MISSING PROOF AND ADDITIONAL THEORETICAL ANALYSIS

## F.1   FACTS USED IN THE PROOF

*Fact* F.1 (Differentiation of integral with a parameter (theorem 2.27 from Folland (1984))). Suppose that $f : X \times [a, b] \to \mathbb{C}(-\infty < a < b < \infty)$ and that $f(\cdot, t) : X \to \mathbb{C}$ is integrable for each $t \in [a, b]$. Let $F(t) = \int_X f(x, t) d\mu(x)$.

[label=.]

1. Suppose that there exists $g \in L^1(\mu)$ such that $|f(x, t)| \le g(x)$ for all $x, t$. If $\lim_{t \to t_0} f(x, t) = f(x, t_0)$ for every $x$, then $\lim_{t \to t_0} F(t) = F(t_0)$; in particular, if $f(x, \cdot)$ is continuous for each $x$, then $F$ is continuous.

2. Suppose that $\partial f / \partial t$ exists and there is a $g \in L^1(\mu)$ such that $|(\partial f / \partial t)(x, t)| \le g(x)$ for all $x, t$. Then $F$ is differentiable and $F'(x) = \int (\partial f / \partial t)(x, t) d\mu(x)$.

*Fact* F.2 (Tower Property). For any random variables $X$ and $Y$, we have

$$\mathbb{E}\left[\mathbb{E}\left[X|Y\right]\right] = \mathbb{E}\left[X\right].$$

*Fact* F.3 (Every point is a fixed point (Khaled & Jin, 2023)). Let $\varphi : \mathbb{R}^d \to \mathbb{R}$ be a convex differentiable function. Then

$$\text{prox}_{\gamma\varphi}(x + \gamma\nabla\varphi(x)) = x, \qquad \forall\gamma > 0, \quad \forall x \in \mathbb{R}^d.$$

In particular, if $x_\star$ is a minimizer of $\varphi$, then $\text{prox}_{\gamma\varphi}(x_\star) = x_\star$.

*Proof.* Evaluating the proximity operator is equivalent to

$$\text{prox}_{\gamma\varphi}(y) = \arg\min_{x \in \mathbb{R}^d} \left(\varphi(x) + \frac{1}{2\gamma}\|x - y\|^2\right).$$

This is a strongly convex minimization problem for any $\gamma > 0$, hence the (necessarily unique) minimizer $x = \text{prox}_{\gamma\varphi}(y)$ of this problem satisfies the first-order optimality condition

$$\nabla\varphi(x) + \frac{1}{\gamma}(x - y) = 0.$$

Solving for $y$, we observe that this holds for $y = x + \gamma\nabla\phi(x)$. Therefore, $x = \text{prox}_{\gamma\varphi}(x + \gamma\nabla\varphi(x))$. $\square$

*Fact* F.4 (Contractivity of the prox (Mishchenko et al., 2022a)). If $\varphi$ is differentiable and $\mu$-strongly convex, then for all $\gamma > 0$ and for any $x, y \in \mathbb{R}^d$ we have

$$\left\|\text{prox}_{\gamma\varphi}(x) - \text{prox}_{\gamma\varphi}(y)\right\|^2 \le \frac{1}{(1 + \gamma\mu)^2}\|x - y\|^2.$$

*Fact* F.5 (Recurrence (Khaled & Jin, 2023, Lemma 1)). Assume that a sequence $\{s_t\}_{t \ge 0}$ of positive real numbers for all $t \ge 0$ satisfies

$$s_{t+1} \le as_t + b,$$

where $0 < a < 1$ and $b \ge 0$. Then the sequence for all $t \ge 0$ satisfies

$$s_t \le a^t s_0 + b\min\left\{t, \frac{1}{1 - a}\right\}.$$

*Proof.* Unrolling the recurrence, we get

$$s_t \le as_{t-1} + b \le a(as_{t-2} + b) + b \le \cdots \le a^t s_0 + b\sum_{i=0}^{t-1} a^i.$$

We can now bound the sum $\sum_{i=0}^{t-1} a^i$ in two different ways. First, since $a < 1$, we get the estimate

$$\sum_{i=0}^{t-1} a^i \le \sum_{i=0}^{t-1} 1 = t.$$

Second, we sum a geometic series

$$\sum_{i=0}^{t-1} a^i \le \sum_{i=0}^{\inf} a^i = \frac{1}{1-a}.$$

Note that either of these bounds can be better. So, we apply the best of these bounds. Substituing the above two bounds gived the target inequality. $\qquad\square$

### F.2   SIMPLIFIED PROOF OF SPPM

We provide a simplified proof of SPPM (Khaled & Jin, 2023) in this section. Using the fact that $x_\star = \text{prox}_{\gamma f_{\xi_t}}(x_\star + \gamma \nabla f_{\xi_t}(x_\star))$ (see Fact F.3) and then applying contraction of the prox (Fact F.4), we get

$$
\begin{aligned}
\|x_{t+1} - x_\star\|^2 &= \left\| \text{prox}_{\gamma f_{\xi_t}} -x_\star \right\|^2 \\
&\stackrel{(Fact\ F.3)}{=} \left\| \text{prox}_{\gamma f_{\xi_t}}(x_t) - \text{prox}_{\gamma f_{\xi_t}}(x_\star + \gamma \nabla f_{\xi_t}(x_\star)) \right\|^2 \\
&\stackrel{(Fact\ F.4)}{\le} \frac{1}{(1+\gamma\mu)^2} \|x_t - (x_\star + \gamma \nabla f_{\xi_t}(x_\star))\|^2 \\
&= \frac{1}{(1+\gamma\mu)^2} \left( \|x_t - x_\star\|^2 - 2\gamma \langle \nabla f_{\xi_t}(x_\star), x_t - x_\star \rangle + \gamma^2 \|\nabla f_{\xi_t}(x_\star)\|^2 \right).
\end{aligned}
$$

Taking expectation on both sides, conditioned on $x_t$, we get

$$
\begin{aligned}
\mathbb{E}\left[ \|x_{t+1} - x_\star\|^2 | x_t \right] &\le \frac{1}{(1+\gamma\mu)^2} \left( \|x_t - x_\star\|^2 - 2\gamma \langle \mathbb{E}\left[ \nabla f_{\xi_t}(x_\star) \right], x_t - x_\star \rangle + \gamma^2 \mathbb{E}\left[ \|\nabla f_{\xi_t}(x_\star)\|^2 \right] \right) \\
&= \frac{1}{(1+\gamma\mu)^2} \left( \|x_t - x_\star\|^2 + \gamma^2 \sigma_\star^2 \right),
\end{aligned}
$$

where we used the fact that $\mathbb{E}\left[ \nabla f_{\xi_t}(x_\star) \right] = \nabla f(x_\star) = 0$ and $\sigma_\star^2 := \mathbb{E}\left[ \|\nabla f_{\xi_t}(x_\star)\|^2 \right]$. Taking expectation again and applying the tower property (Fact F.2), we get

$$\mathbb{E}\left[ \|x_{t+1} - x_\star\|^2 \right] \le \frac{1}{(1+\gamma\mu)^2} \left( \|x_t - x_\star\|^2 + \gamma^2 \sigma_\star^2 \right).$$

It only remains to solve the above recursion. Luckily, that is exactly what Fact F.5 does. In particular, we use it with $s_t = \mathbb{E}\left[ \|x_t - x_\star\|^2 \right], a = \frac{1}{(1+\gamma\mu)^2}$ and $b = \frac{\gamma^2 \sigma_\star^2}{(1+\gamma\mu)^2}$ to get

$$
\begin{aligned}
\mathbb{E}\left[ \|x_t - x_\star\|^2 \right] &\stackrel{(Fact\ F.5)}{\le} \left( \frac{1}{1+\gamma\mu} \right)^{2t} \|x_0 - x_\star\|^2 + \frac{\gamma^2 \sigma_\star^2}{(1+\gamma\mu)^2} \min\left\{ t, \frac{(1+\gamma\mu)^2}{(1+\gamma\mu)^2 - 1} \right\} \\
&\le \left( \frac{1}{1+\gamma\mu} \right)^{2t} \|x_0 - x_\star\|^2 + \frac{\gamma^2 \sigma_\star^2}{(1+\gamma\mu)^2 - 1} \\
&\le \left( \frac{1}{1+\gamma\mu} \right)^{2t} \|x_0 - x_\star\|^2 + \frac{\gamma \sigma_\star^2}{\gamma\mu^2 + 2\mu}.
\end{aligned}
$$

### F.3 MISSING PROOF OF THEOREM 1

We first prove the following useful lemma.

Let $\phi_\xi : \mathbb{R}^d \to \mathbb{R}$ be differentiable functions for almost all $\xi \sim \mathcal{D}$, with $\phi_\xi$ being $\mu_\xi$-strongly convex for almost all $\xi \sim \mathcal{D}$. Further, let $w_\xi$ be positive scalars. Then the function $\phi := \mathbb{E}_{\xi \sim \mathcal{D}}[w_\xi \phi_\xi]$ is $\mu$-strongly convex with $\mu = \mathbb{E}_{\xi \sim \mathcal{D}}[w_\xi \mu_\xi]$.

*Proof.* By assumption,

$$\phi_\xi(y) + \langle \nabla \phi_\xi(y), x - y \rangle + \frac{\mu_\xi}{2}\|x - y\|^2 \le \phi_\xi(x), \quad \text{for almost all } \xi \in \mathcal{D}, \forall x, y \in \mathbb{R}^d.$$

This means that

$$\mathbb{E}_{\xi \sim \mathcal{D}}\left[w_\xi\left(\phi_\xi(y) + \langle \nabla \phi_\xi(y), x - y \rangle + \frac{\mu_\xi}{2}\|x - y\|^2\right)\right] \le \mathbb{E}_{\xi \sim \mathcal{D}}[w_\xi \phi_\xi(x)], \quad \forall x, y \in \mathbb{R}^d,$$

which is equivalent to

$$\phi(y) + \langle \nabla \phi(y), x - y \rangle + \frac{\mathbb{E}_{\xi \sim \mathcal{D}}[w_\xi \mu_\xi]}{2}\|x - y\|^2 \le \phi(x), \quad \forall x, y \in \mathbb{R}^d,$$

So, $\phi$ is $\mu$-strongly convex. $\qquad \square$

Now, we are ready to prove our main Theorem 1.

*Proof.* Let $C$ be any (necessarily nonempty) subset of $[n]$ such that $p_C > 0$. Recall that in view of Equation (8) we have

$$f_C(x) = \mathbb{E}_{\xi \sim \mathcal{D}}\left[\frac{I(\xi \in C)}{p_\xi}f_\xi(x)\right]$$

i.e., $f_C$ is a conic combination of the functions $\{f_\xi : \xi \in C\}$ with weights $w_\xi = \frac{I(\xi \in C)}{p_\xi}$. Since each $f_\xi$ is $\mu_\xi$-strongly convex, Appendix F.3 says that $f_C$ is $\mu_C$-strongly convex with

$$\mu_C := \mathbb{E}_{\xi \sim \mathcal{D}}\left[\frac{I(\xi \in C)\mu_\xi}{p_\xi}\right].$$

So, every such $f_C$ is $\mu$-strongly convex with

$$\mu = \mu_{\text{AS}} := \min_{C \subseteq [n], p_C > 0} \mathbb{E}_{\xi \sim \mathcal{D}}\left[\frac{I(\xi \in C)\mu_\xi}{p_\xi}\right].$$

Further, the quantity $\sigma_\star^2$ from (2.3) is equal to

$$\sigma_\star^2 := \mathbb{E}_{\xi \sim \mathcal{D}}\left[\|\nabla f_\xi(x_\star)\|^2\right] \overset{Eqn. (10)}{=} \sum_{C \subseteq [n], p_C > 0} p_C \|\nabla f_C(x_\star)\|^2 := \sigma_{\star, \text{AS}}^2.$$

Incorporating Appendix F.2 into the above equation, we prove the theorem. $\qquad \square$

### F.4 THEORY FOR EXPECTATION FORMULATION

We will formally define our optimization objective, focusing on minimization in expectation form. We consider

$$\min_{x \in \mathbb{R}^d} f(x) := \mathbb{E}_{\xi \sim \mathcal{D}}[f_\xi(x)], \tag{6}$$

where $f_\xi : \mathbb{R}^d \to \mathbb{R}, \xi \sim \mathcal{D}$ is a random variable following distribution $\mathcal{D}$.

**Assumption F.6.** Function $f_\xi : \mathbb{R}^d \to \mathbb{R}$ is differentiable for almost all samples $\xi \sim \mathcal{D}$.

This implies that $f$ is differentiable. We will implicitly assume that the order of differentiation and expectation can be swapped [1], which means that

$$\nabla f(x) \overset{Eqn.\ (1)}{=} \nabla \mathbb{E}_{\xi \sim \mathcal{D}} [f_\xi(x)] = \mathbb{E}_{\xi \sim \mathcal{D}} [\nabla f_\xi(x)].$$

**Assumption F.7.** Function $f_\xi : \mathbb{R}^d \to \mathbb{R}$ is $\mu$-strongly convex for almost all samples $\xi \sim \mathcal{D}$, where $\mu > 0$. That is

$$f_\xi(y) + \langle \nabla f_\xi, x - y \rangle + \frac{\mu}{2} \|x - y\|^2 \leq f_\xi(x),$$

for all $x, y \in \mathbb{R}^d$.

This implies that $f$ is $\mu$-strongly convex, and hence $f$ has a unique minimizer, which we denote by $x_\star$. We know that $\nabla f(x_\star) = 0$. Notably, we do *not* assume $f$ to be $L$-smooth.

Let $\mathcal{S}$ be a probability distribution over all *finite* subsets of $\mathbb{N}$. Given a random set $S \sim \mathcal{S}$, we define

$$p_i := \mathrm{Prob}(i \in S), \quad i \in \mathbb{N}.$$

We will restrict our attention to proper and nonvacuous random sets.

**Assumption F.8.** $S$ is proper (i.e., $p_i > 0$ for all $i \in \mathbb{N}$) and nonvacuous (i.e., $\mathrm{Prob}(S = \emptyset) = 0$).

Let $C$ be the selected cohort. Given $\emptyset \neq C \subset \mathbb{N}$ and $i \in \mathbb{N}$, we define

$$v_i(C) := \begin{cases} \frac{1}{p_i} & i \in C \\ 0 & i \notin C, \end{cases} \tag{7}$$

and

$$f_C(x) := \mathbb{E}_{\xi \sim \mathcal{D}} [v_\xi(C) f_\xi(x)] \overset{Eqn.\ (7)}{=} \mathbb{E}_{\xi \sim \mathcal{D}} \left[ \frac{I(\xi \in C)}{p_\xi} f_\xi(x) \right]. \tag{8}$$

Note that $v_i(S)$ is a random variable and $f_S$ is a random function. By construction, $\mathbb{E}_{S \sim \mathcal{S}} [v_i(S)] = 1$ for all $i \in \mathbb{N}$, and hence

$$\begin{aligned} \mathbb{E}_{S \sim \mathcal{S}} [f_S(x)] &= \mathbb{E}_{S \sim \mathcal{S}} [\mathbb{E}_{\xi \sim \mathcal{D}} [v_\xi(C) \nabla f_\xi(x)]] \\ &= \mathbb{E}_{\xi \sim \mathcal{D}} [\mathbb{E}_{S \sim \mathcal{S}} [v_\xi(S)] \nabla f_\xi(x)] = \mathbb{E}_{\xi \sim \mathcal{D}} [f_\xi(x)] = f(x). \end{aligned}$$

Therefore, the optimization problem in Equation (1) is equivalent to the stochastic optimization problem

$$\min_{x \in \mathbb{R}^d} \{f(x) := \mathbb{E}_{S \sim \mathcal{S}} [f_S(x)]\}. \tag{9}$$

Further, if for each $C \subset \mathbb{N}$ we let $p_C := \mathrm{Prob}(S = C)$, $f$ can be written in the equivalent form

$$f(x) = \mathbb{E}_{S \sim \mathcal{S}} [f_S(x)] = \sum_{C \subset \mathbb{N}} p_C f_C(x) = \sum_{C \subset \mathbb{N}, p_C > 0} p_C f_C(x). \tag{10}$$

**Theorem F.9** (Main Theorem). *Let Assumption 2.1 (diferentiability) and Assumption 2.2 (strong convexity) hold. Let $S$ be a random set satisfying Assumption 2.3, and define*

$$\mu_{\mathrm{AS}} := \min_{C \subset \mathbb{N}, p_C > 0} \mathbb{E}_{\xi \sim \mathcal{D}} \left[ \frac{I(\xi \in C) \mu_\xi}{p_\xi} \right],$$

$$\sigma_{\star,\mathrm{AS}}^2 := \sum_{C \subset \mathbb{N}, p_C > 0} p_C \|\nabla f_C(x_\star)\|^2. \tag{11}$$

*Let $x_0 \in \mathbb{R}^d$ be an arbitrary starting point. Then for any $t \geq 0$ and any $\gamma > 0$, the iterates of* SPPM-AS *(Algorithm 1) satisfy*

$$\mathbb{E} [\|x_t - x_\star\|^2] \leq \left( \frac{1}{1 + \gamma \mu_{\mathrm{AS}}} \right)^{2t} \|x_0 - x_\star\|^2 + \frac{\gamma \sigma_{\star,\mathrm{AS}}^2}{\gamma \mu_{\mathrm{AS}}^2 + 2\mu_{\mathrm{AS}}}.$$

---

[1]This assumption satisfies the conditions required for the theorem about differentiating an integral with a parameter (Fact F.1).

## F.5 MISSING PROOF OF ITERATION COMPLEXITY OF SPPM-AS

We have seen above that accuracy arbitrarily close to (but not reaching) $\sigma^2_{\star,\text{AS}}/\mu^2_{\text{AS}}$ can be achieved via a single step of the method, provided the stepsize $\gamma$ is large enough. Assume now that we aim for $\epsilon$ accuracy where $\epsilon \leq \sigma^2_{\star,\text{AS}}/\mu^2_{\text{AS}}$. Using the inequality $1 - k \leq \exp(-k)$ which holds for all $k > 0$, we get

$$\left(\frac{1}{1+\gamma\mu_{\text{AS}}}\right)^{2t} = \left(1 - \frac{\gamma\mu}{1+\gamma\mu_{\text{AS}}}\right)^{2t} \leq \exp\left(-\frac{2\gamma\mu_{\text{AS}}t}{1+\gamma\mu_{\text{AS}}}\right)$$

Therefore, provided that

$$t \geq \frac{1+\gamma\mu_{\text{AS}}}{2\gamma\mu_{\text{AS}}} \log\left(\frac{2\left\|x_0 - x_\star\right\|^2}{\varepsilon}\right),$$

we get $\left(\frac{1}{1+\gamma\mu_{\text{AS}}}\right)^{2t}\left\|x_0 - x_\star\right\|^2 \leq \frac{\varepsilon}{2}$. Furthermore, as long as $\gamma \leq \frac{2\varepsilon\mu_{\text{AS}}}{2\sigma^2_{\star,\text{AS}}-\varepsilon\mu^2_{\text{AS}}}$ (this is true provided that the more restrictive but also more elegant-looking condition $\gamma \leq \varepsilon\mu_{\text{AS}}/\sigma^2_{\star,\text{AS}}$ holds), we get $\frac{\gamma\sigma^2_{\star,\text{AS}}}{\gamma\mu^2_{\text{AS}}+2\mu_{\text{AS}}} \leq \frac{\varepsilon}{2}$. Putting these observations together, we conclude that with the stepsize $\gamma = \varepsilon\mu_{\text{AS}}/\sigma^2_{\star,\text{AS}}$, we get $\text{E}\left[\left\|x_t - x_\star\right\|^2\right] \leq \varepsilon$ provided that

$$t \geq \frac{1+\gamma\mu_{\text{AS}}}{2\gamma\mu_{\text{AS}}} \log\frac{2\left\|x_0 - x_\star\right\|^2}{\varepsilon} = \left(\frac{\sigma^2_{\star,\text{AS}}}{2\varepsilon\mu^2_{\text{AS}}} + \frac{1}{2}\right) \log\left(\frac{2\left\|x_0 - x_\star\right\|^2}{\varepsilon}\right).$$

## F.6 $\sigma^2_{\star,\text{NICE}}(\tau)$ AND $\mu_{\text{NICE}}(\tau)$ ARE MONOTONOUS FUNCTIONS OF $\tau$

For all $0 \leq \tau \leq n-1$:

1. $\mu_{\text{NICE}}(\tau+1) \geq \mu_{\text{NICE}}(\tau)$,
2. $\sigma^2_{\star,\text{NICE}}(\tau) = \frac{\frac{n}{\tau}-1}{n-1}\sigma^2_{\star,\text{NICE}}(1) \leq \frac{1}{\tau}\sigma^2_{\star,\text{NICE}}(1)$.

*Proof.*      1. Pick any $1 \leq \tau < n$, and consider a set $C$ for which the minimum is attained in

$$\mu_{\text{NICE}}(\tau+1) = \min_{C\subseteq[n],|C|=\tau+1} \frac{1}{\tau+1}\sum_{i\in C}\mu_i.$$

Let $j = \arg\max_{i\in C}\mu_i$. That is, $\mu_j \geq \mu_i$ for all $i \in C$. Let $C_j$ be the set obtained from $C$ by removing the element $j$. Then $|C_j| = \tau$ and

$$\mu_j = \max_{i\in C}\mu_i \geq \max_{i\in C_j}\mu_i \geq \frac{1}{\tau}\sum_{i\in C_j}\mu_i.$$

By adding $\sum_{i\in C_j}\mu_i$ to the above inequality, we obtain

$$\mu_j + \sum_{i\in C_j}\mu_i \geq \frac{1}{\tau}\sum_{i\in C_j}\mu_i + \sum_{i\in C_j}\mu_i.$$

Observe that the left-hand side is equal to $\sum_{i\in C}\mu_i$, and the right-hand side is equal to $\frac{\tau+1}{\tau}\sum_{i\in C_j}\mu_i$. If we divide both sides by $\tau + 1$, we obtain

$$\frac{1}{\tau+1}\sum_{i\in C}\mu_i \geq \frac{1}{\tau}\sum_{i\in C_j}\mu_i.$$

Since the left-hand side is equal to $\mu_{\text{NICE}}(\tau+1)$, and the right hand side is an upper bound on $\mu_{\text{NICE}}(\tau)$, we conclude that $\mu_{\text{NICE}}(\tau+1) \geq \mu_{\text{NICE}}(\tau)$.

2. In view of (8) we have

$$f_C(x) = \sum_{i \in C} \frac{1}{np_i} f_i(x).$$

(12)

$$\sigma_{\star,\mathrm{AS}}^2 \quad = \quad \mathrm{E}_{S \sim \mathcal{S}} \left[ \left\| \sum_{i \in S} \frac{1}{np_i} \nabla f_i(x_\star) \right\|^2 \right] = \mathrm{E}_{S \sim \mathcal{S}} \left[ \left\| \sum_{i \in S} \frac{1}{\tau} \nabla f_i(x_\star) \right\|^2 \right]$$

(13)

Let $\chi_i$ be the random variable defined by

$$\chi_j = \begin{cases} 1 & j \in S \\ 0 & j \notin S. \end{cases}$$

(14)

It is easy to show that

$$\mathbb{E}[\chi_j] = \mathrm{Prob}(j \in S) = \frac{\tau}{n}.$$

(15)

Let fix the cohort S. Let $\chi_{ij}$ be the random variable defined by

$$\chi_{ij} = \begin{cases} 1 & i \in S \text{ and } j \in S \\ 0 & \text{otherwise}. \end{cases}$$

(16)

Note that

$$\chi_{ij} = \chi_i \chi_j.$$

(17)

Further, it is easy to show that

$$\mathbb{E}[\chi_{ij}] = \mathrm{Prob}(i \in S, j \in S) = \frac{\tau(\tau - 1)}{n(n - 1)}.$$

(18)

Denote $a_i := \nabla f_i(x_\star)$.

$$
\begin{aligned}
\mathbb{E}\left[\left\|\frac{1}{\tau}\sum_{i\in S}a_i\right\|^2\right] &= \frac{1}{\tau^2}\mathbb{E}\left[\left\|\sum_{i\in S}a_i\right\|^2\right] \\
&= \frac{1}{\tau^2}\mathbb{E}\left[\left\|\sum_{i=1}^n \chi_i a_i\right\|^2\right] \\
&= \frac{1}{\tau^2}\mathbb{E}\left[\sum_{i=1}^n \|\chi_i a_i\|^2 + \sum_{i\neq j}\langle \chi_i a_i, \chi_j a_j\rangle\right] \\
&= \frac{1}{\tau^2}\mathbb{E}\left[\sum_{i=1}^n \|\chi_i a_i\|^2 + \sum_{i\neq j}\chi_{ij}\langle a_i, a_j\rangle\right] \\
&= \frac{1}{\tau^2}\sum_{i=1}^n \mathbb{E}[\chi_i]\|a_i\|^2 + \sum_{i\neq j}\mathbb{E}[\chi_{ij}]\langle a_i, a_j\rangle \\
&= \frac{1}{\tau^2}\left(\frac{\tau}{n}\sum_{i=1}^n \|a_i\|^2 + \frac{\tau(\tau-1)}{n(n-1)}\sum_{i\neq j}\langle a_i, a_j\rangle\right) \\
&= \frac{1}{\tau n}\sum_{i=1}^n \|a_i\|^2 + \frac{\tau-1}{\tau n(n-1)}\sum_{i\neq j}\langle a_i, a_j\rangle \\
&= \frac{1}{\tau n}\sum_{i=1}^n \|a_i\|^2 + \frac{\tau-1}{\tau n(n-1)}\left(\left\|\sum_{i=1}^n a_j\right\|^2 - \sum_{i=1}^n \|a_i\|^2\right) \\
&= \frac{n-\tau}{\tau(n-1)}\frac{1}{n}\sum_{i=1}^n \|a_i\|^2 + \frac{n(\tau-1)}{\tau(n-1)}\left\|\frac{1}{n}\sum_{i=1}^n a_i\right\|^2 \\
&= \frac{n-\tau}{\tau(n-1)}\frac{1}{n}\sum_{i=1}^n \|\nabla f_i(x_\star)\|^2 + \frac{n(\tau-1)}{\tau(n-1)}\left\|\frac{1}{n}\sum_{i=1}^n \nabla f_i(x_\star)\right\|^2 \\
&= \frac{n-\tau}{\tau(n-1)}\frac{1}{n}\sum_{i=1}^n \|\nabla f_i(x_\star)\|^2 \\
&\leq \frac{1}{\tau}\frac{1}{n}\sum_{i=1}^n \|\nabla f_i(x_\star)\|^2
\end{aligned}
$$

$\square$

### F.7 MISSING PROOF OF LEMMA 1

For ease of notation, let $a_i = \nabla f_i(x_\star)$ and $\hat{z}_j = |C_j|\, a_{\xi_j}$, and recall that

$$
\sigma_{\star,\mathrm{SS}}^2 = \mathrm{E}_{\xi_1,\ldots,\xi_b}\left[\left\|\frac{1}{n}\sum_{j=1}^b \hat{z}_j\right\|^2\right]. \tag{19}
$$

where $\xi_j \in C_j$ is chosen uniformly at random. Further, for each $j \in [b]$, let $z_j = \sum_{i \in C_j} a_i$. Observe that $\sum_{j=1}^b z_j = \sum_{j=1}^b \sum_{i \in C_j} a_i = \sum_{i=1}^n a_i = \nabla f(x_\star) = 0$. Therefore,

$$
\left\| \frac{1}{n} \sum_{j=1}^b \hat{z}_j \right\|^2 = \frac{1}{n^2} \left\| \sum_{j=1}^b \hat{z}_j - \sum_{j=1}^b z_j \right\|^2
$$

$$
= \frac{b^2}{n^2} \left\| \frac{1}{b} \sum_{j=1}^b (\hat{z}_j - z_j) \right\|^2
$$

$$
\leq \frac{b^2}{n^2} \frac{1}{b} \sum_{j=1}^b \| \hat{z}_j - z_j \|^2
$$

$$
= \frac{b}{n^2} \sum_{j=1}^b \| \hat{z}_j - z_j \|^2 , \tag{20}
$$

where the inequality follows from convexity of the function $u \mapsto \|u\|^2$. Next,

$$
\| \hat{z}_j - z_j \|^2 = \left\| |C_j| \, a_{\xi_j} - \sum_{i \in C_j} a_i \right\|^2 = |C_j|^2 \left\| a_{\xi_j} - \frac{1}{|C_j|} \sum_{i \in C_j} a_i \right\|^2 \leq |C_j|^2 \sigma_j^2. \tag{21}
$$

By combining Equation (19), Equation (20) and Equation (21), we get

$$
\sigma_{\star,\mathrm{SS}}^2 \overset{Eqn.\ (19)}{=} \mathrm{E}_{\xi_1,\ldots,\xi_b} \left[ \left\| \frac{1}{n} \sum_{j=1}^b \hat{z}_j \right\|^2 \right]
$$

$$
\overset{Eqn.\ (20)}{\leq} \mathrm{E}_{\xi_1,\ldots,\xi_b} \left[ \frac{b}{n^2} \sum_{j=1}^b \| \hat{z}_j - z_j \|^2 \right]
$$

$$
\overset{Eqn.\ (21)}{\leq} \mathrm{E}_{\xi_1,\ldots,\xi_b} \left[ \frac{b}{n^2} \sum_{j=1}^b |C_j|^2 \sigma_j^2 \right]
$$

$$
= \frac{b}{n^2} \sum_{j=1}^b |C_j|^2 \sigma_j^2.
$$

The last expression can be further bounded as follows:

$$
\frac{b}{n^2} \sum_{j=1}^b |C_j|^2 \sigma_j^2 \leq \frac{b}{n^2} \left( \sum_{j=1}^b |C_j|^2 \right) \max_j \sigma_j^2 \leq \frac{b}{n^2} \left( \sum_{j=1}^b |C_j| \right)^2 \max_j \sigma_j^2 = b \max_j \sigma_j^2,
$$

where the second inequality follows from the relation $\|u\|_2 \leq \|u\|_1$ between the $L_2$ and $L_1$ norms, and the last identity follows from the fact that $\sum_{j=1}^b |C_j| = n$.

## F.8 STRATIFIED SAMPLING AGAINST BLOCK SAMPLING AND NICE SAMPLING

In this section, we present a theoretical comparison of block sampling and its counterparts, providing a theoretical justification for selecting block sampling as the default clustering method in future experiments. Additionally, we compare various sampling methods, all with the same sampling size, $b$: $b$-nice sampling, block sampling with $b$ clusters, and block sampling, where all clusters are of uniform size $b$.

**Assumption F.10.** For simplicity of comparison, we assume $b$ clusters, each of the same size, $b$:

$$
|C_1| = |C_2| = \ldots = |C_b| = b.
$$

It is crucial to acknowledge that, without specific assumptions, the comparison of different sampling methods may not provide meaningful insights. For instance, the scenario described in Lemma 1, characterized by complete inter-cluster homogeneity, demonstrates that stratified sampling achieves a variance term, denoted as $\sigma_{\star,\mathrm{SS}}^2$, which is lower than the variance terms associated with both block sampling and nice sampling. However, a subsequent example illustrates examples in which the variance term for block sampling surpasses those of block sampling and nice sampling. Without imposing any additional clustering assumptions, there exist examples for any arbitrary $n$, such that $\sigma_{\star,\mathrm{SS}}^2 \geq \sigma_{\star,\mathrm{BS}}^2$ and $\sigma_{\star,\mathrm{SS}}^2 \geq \sigma_{\star,\mathrm{NICE}}^2$.

*Proof.* **Counterexample when SS is worse in neighborhood than BS**
Assume we have such clustering and $\nabla f_i(x_\star)$ such that the centroids of each cluster are equal to zero: $\forall i \in [b]$, $\frac{1}{|C_i|} \sum_{j \in C_i} \nabla f_j(x_\star) = 0$. For instance, this can be achieved in the following case: The dimension is $d = 2$, all clusters are of equal size $m$, then assign $\forall i \in [b]$, $\forall j \in C_i$, $\nabla f_j(x_\star) = \left( Re\left(\omega^{mj+i}\right), Im\left(\omega^{mj+i}\right) \right)$ where $\omega = \sqrt[n]{1} \in \mathbb{C}$. Let us calculate $\sigma_{\star,\mathrm{BS}}^2$:

$$\sigma_{\star,\mathrm{BS}}^2 := \sum_{j=1}^{b} q_j \left\| \sum_{i \in C_j} \frac{1}{np_i} \nabla f_i(x_\star) \right\|^2 =$$

$$= \frac{1}{n^2} \sum_{j=1}^{b} \frac{|C_j|^2}{q_j} \left\| \frac{1}{|C_j|} \sum_{i \in C_j} \nabla f_i(x_\star) \right\|^2 = 0.$$

As a result:

$$\sigma_{\star,\mathrm{BS}}^2 = 0 \leq \sigma_{\star,\mathrm{SS}}^2.$$

**Counterexample when SS is worse in neighborhood than NICE**
Here, we employ a similar proof technique as in the proof of Lemma 2. Let us choose such clustering $\mathcal{C}_{b,\mathrm{SS,max}} = \arg\max_{\mathcal{C}_b} \sigma_{\star,\mathrm{SS}}^2(\mathcal{C}_b)$. Denote $\mathbf{i}_b := (i_1, \cdots, i_b)$, $\mathbf{C}_b := C_1 \times \cdots \times C_b$, and $S_{\mathbf{i}_b} := \left\| \frac{1}{\tau} \sum_{i \in \mathbf{i}_b} \nabla f_i(x_\star) \right\|$.

$$\sigma_{\star,\mathrm{NICE}}^2 = \frac{1}{C(n,\tau)} \sum_{C \subseteq [n], |C| = \tau} \left\| \frac{1}{\tau} \sum_{i \in C} \nabla f_i(x_\star) \right\|^2$$

$$= \frac{1}{C(n,b)} \sum_{\mathbf{i}_b \subseteq [n]} S_{\mathbf{i}_b}$$

$$\overset{1}{=} \frac{1}{\#\text{clusterizations}} \sum_{\mathcal{C}_b} \frac{1}{b^b} \sum_{\mathbf{i}_b \in \mathbf{C}_b} S_{\mathbf{i}_b}$$

$$= \frac{1}{\#\text{clusterizations}} \sum_{\mathcal{C}_b} \sigma_{\star,\mathrm{SS}}^2(\mathcal{C}_b)$$

$$\overset{2}{\leq} \sigma_{\star,\mathrm{SS}}^2(\mathcal{C}_{b,\mathrm{SS,max}}).$$

Equation 1 holds because, in every clusterization $\mathcal{C}_b$, there are $\frac{1}{b^b}$ possible sample combinations $\mathbf{i}_b$. Due to symmetry, one can conclude that each combination $S_{\mathbf{i}_b}$ is counted the same number of times. Equation 2 follows from the definition of $\mathcal{C}_{b,\mathrm{SS,max}}$.
For illustrative purposes, we can demonstrate this effect with a specific example. Let $n = 4$ and define $\forall i \, a_i = \nabla f_i(x^*) \in \mathbb{R}^2$. Let $a_1 = (0,1)^T$, $a_2 = (1,0)^T$, $a_3 = (0,-1)^T$, and $a_4 = (-1,0)^T$. Then fix clustering $\mathcal{C}_b = \{C_1 = \{a_1, a_3\}, C_2 = \{a_2, a_4\}\}$. Then:

$$\sigma_{\star,\mathrm{SS}}^2 = \frac{1}{4} \sum_{\mathbf{i}_b \in \mathcal{C}_b} \left\| \frac{a_{i_1} + a_{i_2}}{2} \right\|^2$$

$$= \frac{1}{4} \sum_{\mathbf{i}_b \in \mathcal{C}_b} \left\| (\pm\frac{1}{2}, \pm\frac{1}{2}) \right\|^2$$

$$= \frac{1}{2}.$$

$$\sigma^2_{\star,\text{NICE}} = \frac{1}{C(4,2)} \sum_{i<j} \left\| \frac{a_i + a_j}{2} \right\|^2$$

$$= \frac{1}{6} \sum_{i<j} \left\| \frac{a_i + a_j}{2} \right\|^2$$

$$= \frac{1}{6} \left( \left[ \left\| \frac{a_1 + a_3}{2} \right\|^2 + \left\| \frac{a_2 + a_4}{2} \right\|^2 \right] + 2 \times \left\| \frac{a_{i_1} + a_{i_2}}{2} \right\|^2 \right)$$

$$= \frac{1}{6} \left( 0 + 2 \times 2 \times \frac{1}{2} \right)$$

$$= \frac{1}{3}$$

$$= \frac{2}{3} \times \sigma^2_{\star,\text{SS}}$$

$$\leq \sigma^2_{\star,\text{SS}}$$

$\square$

To select the optimal clustering, we will choose the clustering that minimizes $\sigma^2_{\star,\text{SS}}$.

**Definition F.11** (Stratified sampling optimal clustering). Denote the clustering of workers into blocks as $\mathcal{C}_b := \{C_1, C_2, \ldots, C_b\}$, such that the disjoint union of all clusters $C_1 \cup C_2 \cup \ldots \cup C_b = [n]$. Define *block sampling Optimal Clustering* as the clustering configuration that minimizes $\sigma^2_{\star,\text{SS}}$, formally given by:

$$\mathcal{C}_{b,\text{SS}} := \arg\min_{\mathcal{C}_b} \sigma^2_{\star,\text{SS}}(\mathcal{C}_b).$$

Proof of the Lemma 2 is provided below.

*Proof.*    1. Denote $\mathbf{i}_b := (i_1, \cdots, i_b)$, $\mathbf{C}_b := C_1 \times \cdots \times C_b$, and $S_{\mathbf{i}_b} := \left\| \frac{1}{\tau} \sum_{i \in \mathbf{i}_b} \nabla f_i(x_\star) \right\|$.

$$\sigma^2_{\star,\text{NICE}} = \frac{1}{C(n,\tau)} \sum_{C \subseteq [n], |C| = \tau} \left\| \frac{1}{\tau} \sum_{i \in C} \nabla f_i(x_\star) \right\|^2$$

$$= \frac{1}{C(n,b)} \sum_{\mathbf{i}_b \subseteq [n]} S_{\mathbf{i}_b}$$

$$\overset{1}{=} \frac{1}{\#\text{clusterizations}} \sum_{\mathcal{C}_b} \frac{1}{b^b} \sum_{\mathbf{i}_b \in \mathbf{C}_b} S_{\mathbf{i}_b}$$

$$= \frac{1}{\#\text{clusterizations}} \sum_{\mathcal{C}_b} \sigma^2_{\star,\text{SS}}(\mathcal{C}_b)$$

$$\overset{2}{\geq} \sigma^2_{\star,\text{SS}}(\mathcal{C}_{b,\text{SS,min}})$$

Equation 1 holds because, in every clusterization $\mathcal{C}_b$, there are $\frac{1}{b^b}$ possible sample combinations $\mathbf{i}_b$. Due to symmetry, one can conclude that each combination $S_{\mathbf{i}_b}$ is counted the same number of times. Equation 2 follows from the definition of $\mathcal{C}_{b,\text{SS,min}}$ as the clustering that minimizes $\sigma^2_{\star,\text{SS}}$, according to Definition F.11.

2. The neighborhood size for SPPM-AS is given by $\frac{\gamma\sigma^2_{\star,\mathrm{AS}}}{\gamma\mu^2_{\mathrm{AS}}+2\mu_{\mathrm{AS}}}$, denoted as $U_{\mathrm{AS}}$ for simplicity. Define:

$$\mu_{\mathrm{NICE}(b)} := \min_{\substack{C \subseteq [n] \\ |C|=b}} \frac{1}{b} \sum_{i \in C} \mu_i,$$

$$\mu_{\mathrm{SS}} := \min_{\mathbf{i}_b \in \mathbf{C}_b} \sum_{j=1}^{b} \frac{\mu_{i_j}|C_j|}{n} \overset{\text{Asm. 10}}{=} \min_{\mathbf{i}_b \in \mathbf{C}_b} \sum_{j=1}^{b} \frac{\mu_{i_j}b}{b^2} = \min_{\mathbf{i}_b \in \mathbf{C}_b} \frac{1}{b} \sum_{j=1}^{b} \mu_{i_j}.$$

Using the definition of the set $\mathbf{C}_b := C_1 \times C_2 \times \cdots \times C_b$, we have $\mathbf{C}_b \subseteq \{C \subseteq [n] \mid |C| = b\}$. Applying this fact, we obtain:

$$\mu_{\mathrm{SS}} = \min_{\mathbf{i}_b \in \mathbf{C}_b} \frac{1}{b} \sum_{j \in \mathbf{i}_b} \mu_j \geq \mu_{\mathrm{NICE}(b)}.$$

Combining the above with $\sigma^2_{\star,\mathrm{SS}}(\mathcal{C}_{b,\mathrm{SS}}) \leq \sigma^2_{\star,\mathrm{NICE}}$, we obtain that $U_{\mathrm{SS}}(\mathcal{C}_{b,\mathrm{SS}}) \leq U_{\mathrm{NICE}}$, demonstrating the variance reduction of SS compared to NICE.

$\square$

Consider the number of clusters and the size of each cluster, with $b = 2$, under Assumption F.10. Then, $\sigma^2_{\star,\mathrm{SS}}(\mathcal{C}_{b,\mathrm{SS}}) \leq \sigma^2_{\star,\mathrm{BS}}$.

*Proof.* Let $n = 4$, $b = 2$. Denote $\forall i\ a_i = \nabla f_i(x_\star)$. Define $S^2 := \sum_{i<j} \left\| \frac{a_i+a_j}{2} \right\|^2$.

$$\sigma^2_{\star,\mathrm{SS}} = \frac{1}{4} \left( S^2 - \left\| \frac{a_{C_1^1} + a_{C_1^2}}{2} \right\|^2 - \left\| \frac{a_{C_2^1} + a_{C_2^2}}{2} \right\|^2 \right)$$

$$= \frac{1}{4} \left( S^2 - 2\sigma^2_{\star,\mathrm{BS}} \right)$$

$\mathcal{C}_{b,\mathrm{SS}}$ clustering minimizes $\sigma^2_{\star,\mathrm{SS}}$, thereby maximizing $\sigma^2_{\star,\mathrm{BS}}$. Thus,

$$\sigma^2_{\star,\mathrm{SS}} = \frac{1}{4} \left( \left[ \left\| \frac{a_{C_1^1} + a_{C_2^1}}{2} \right\|^2 + \left\| \frac{a_{C_1^2} + a_{C_2^2}}{2} \right\|^2 \right] + \left[ \left\| \frac{a_{C_1^1} + a_{C_2^2}}{2} \right\|^2 + \left\| \frac{a_{C_1^2} + a_{C_2^1}}{2} \right\|^2 \right] \right)$$

$$= \frac{1}{4} \left( 2\sigma^2_{\star,\mathrm{BS}}\left((C_1^1,C_2^1),(C_1^2,C_2^2)\right) + 2\sigma^2_{\star,\mathrm{BS}}\left((C_1^1,C_2^2),(C_1^2,C_2^1)\right) \right)$$

$$= \frac{1}{2} \left( \sigma^2_{\star,\mathrm{BS}}\left((C_1^1,C_2^1),(C_1^2,C_2^2)\right) + \sigma^2_{\star,\mathrm{BS}}\left((C_1^1,C_2^2),(C_1^2,C_2^1)\right) \right)$$

$$\leq \sigma^2_{\star,\mathrm{BS}}.$$

$\square$

However, it is possible that this relationship might hold more generally. Empirical experiments for different configurations, such as $b = 3$, support this possibility. For example, with $n = 9$, $b = 3$, and $d = 10$, Python simulations where gradients $\nabla f_i$ are sampled from $\mathcal{N}(0,1)$ and $\mathcal{N}(e,1)$ across 1000 independent trials, show that $\sigma^2_{\star,\mathrm{SS}} \leq \sigma^2_{\star,\mathrm{BS}}$. Question of finding theoretical proof for arbitrary $n$ remains open and has yet to be addressed in the existing literature.

## F.9 DIFFERENT APPROACHES OF FEDERATED AVERAGING

Proof of Theorem 2:

*Proof.*

$$\|x_t - x_\star\|^2 = \left\| \sum_{i \in S_t} \frac{1}{|S_t|} \operatorname{prox}_{\gamma f_i}(x_{t-1}) - \frac{1}{|S_t|} \sum_{i \in S_t} x_\star \right\|^2$$

$$\stackrel{(Fact\ F.3)}{=} \left\| \sum_{i \in S_t} \frac{1}{|S_t|} \left[ \operatorname{prox}_{\gamma f_i}(x_{t-1}) - \operatorname{prox}_{\gamma f_i}(x_\star + \gamma \nabla f_i(x_\star)) \right] \right\|^2$$

$$\stackrel{Jensen}{\leq} \sum_{i \in S_t} \frac{1}{|S_t|} \left\| \left[ \operatorname{prox}_{\gamma f_i}(x_{t-1}) - \operatorname{prox}_{\gamma f_i}(x_\star + \gamma \nabla f_i(x_\star)) \right] \right\|^2$$

$$\stackrel{(Fact\ F.4)}{\leq} \sum_{i \in S_t} \frac{1}{|S_t|} \frac{1}{(1 + \gamma \mu_i)^2} \|x_{t-1} - (x_\star + \gamma \nabla f_i(x_\star))\|^2$$

$$\mathbb{E}_{S_t \sim \mathcal{S}} \left[ \|x_t - x_\star\|^2 | x_{t-1} \right]$$

$$\leq \mathbb{E}_{S_t \sim \mathcal{S}} \left[ \sum_{i \in S_t} \frac{1}{|S_t|} \frac{1}{(1 + \gamma \mu_i)^2} \|(x_{t-1} - x_\star) - \gamma \nabla f_i(x_\star)\|^2 | x_{t-1} \right]$$

$$\stackrel{Young,\ \alpha_i > 0}{\leq} \mathbb{E}_{S_t \sim \mathcal{S}} \left[ \sum_{i \in S_t} \frac{1}{|S_t|} \frac{1}{(1 + \gamma \mu_i)^2} \left( (1 + \alpha_i) \|x_{t-1} - x_\star\|^2 + \left(1 + \alpha_i^{-1}\right) \|\gamma \nabla f_i(x_\star)\|^2 \right) | x_{t-1} \right]$$

$$\stackrel{\alpha_i = \gamma \mu_i}{=} \mathbb{E}_{S_t \sim \mathcal{S}} \left[ \sum_{i \in S_t} \frac{1}{|S_t|} \frac{1}{(1 + \gamma \mu_i)^2} \left( (1 + \gamma \mu_i) \|x_{t-1} - x_\star\|^2 + \left(1 + \frac{1}{\gamma \mu_i}\right) \|\gamma \nabla f_i(x_\star)\|^2 \right) | x_{t-1} \right]$$

$$= \mathbb{E}_{S_t \sim \mathcal{S}} \left[ \sum_{i \in S_t} \frac{1}{|S_t|} \left( \frac{1}{1 + \gamma \mu_i} \|x_{t-1} - x_\star\|^2 + \frac{\gamma}{(1 + \gamma \mu_i) \mu_i} \|\nabla f_i(x_\star)\|^2 \right) | x_{t-1} \right]$$

$$= \mathbb{E}_{S_t \sim \mathcal{S}} \left[ \frac{1}{|S_t|} \sum_{i \in S_t} \frac{1}{1 + \gamma \mu_i} | x_{t-1} \right] \|x_{t-1} - x_\star\|^2 + \mathbb{E}_{S_t \sim \mathcal{S}} \left[ \frac{1}{|S_t|} \sum_{i \in S_t} \frac{\gamma}{(1 + \gamma \mu_i) \mu_i} \|\nabla f_i(x_\star)\|^2 | x_{t-1} \right]$$

By applying tower property one can get the following:

$$\mathbb{E}_{S_t \sim \mathcal{S}} \left[ \|x_t - x_\star\|^2 \right]$$

$$= \mathbb{E}_{S_t \sim \mathcal{S}} \left[ \frac{1}{|S_t|} \sum_{i \in S_t} \frac{1}{1 + \gamma \mu_i} \right] \|x_{t-1} - x_\star\|^2 + \mathbb{E}_{S_t \sim \mathcal{S}} \left[ \frac{1}{|S_t|} \sum_{i \in S_t} \frac{\gamma}{(1 + \gamma \mu_i) \mu_i} \|\nabla f_i(x_\star)\|^2 \right]$$

$$= A_{\mathcal{S}} \|x_{t-1} - x_\star\|^2 + B_{\mathcal{S}}.$$

where $A_{\mathcal{S}} := \mathbb{E}_{S_t \sim \mathcal{S}} \left[ \frac{1}{|S_t|} \sum_{i \in S_t} \frac{1}{1 + \gamma \mu_i} \right]$ and $B_{\mathcal{S}} := \mathbb{E}_{S_t \sim \mathcal{S}} \left[ \frac{1}{|S_t|} \sum_{i \in S_t} \frac{\gamma}{(1 + \gamma \mu_i) \mu_i} \|\nabla f_i(x_\star)\|^2 \right]$.
By directly applying Fact F.5:

$$\mathbb{E}_{S_t \sim \mathcal{S}} \left[ \|x_t - x_\star\|^2 \right] \leq A_{\mathcal{S}}^t \|x_0 - x_\star\|^2 + \frac{B_{\mathcal{S}}}{1 - A_{\mathcal{S}}}.$$

□

[Inexact formulation of SPPM-AS] Let $b > 0 \in \mathbb{R}$ and define $\widetilde{\operatorname{prox}}_{\gamma f}(x)$ such that $\forall x \left\| \widetilde{\operatorname{prox}}_{\gamma f}(x) - \operatorname{prox}_{\gamma f}(x) \right\|^2 \leq b$. Let Assumption 2.1 and Assumption 2.2 hold. Let $x_0 \in \mathbb{R}^d$ be an arbitrary starting point. Then for any $t \geq 0$ and any $\gamma > 0$, $s > 0$, the iterates of SPPM-AS satisfy

$$\mathbb{E} \left[ \|x_t - x_\star\|^2 \right] \leq \left( \frac{1 + s}{(1 + \gamma \mu)^2} \right)^t \|x_0 - x_\star\|^2 + \frac{(1 + s) \left( \gamma^2 \sigma_\star^2 + s^{-1} b (1 + \gamma \mu)^2 \right)}{\gamma^2 \mu^2 + 2 \gamma \mu - s}.$$

*Proof of Lemma F.9.* We provide more general version of SPPM proof

$$\|x_{t+1} - x_\star\|^2 = \left\|\widetilde{\mathrm{prox}}_{\gamma f_{\xi_t}(x_t)} - \mathrm{prox}_{\gamma f_{\xi_t}}(x_t) + \mathrm{prox}_{\gamma f_{\xi_t}}(x_t) - x_\star\right\|^2$$

$$\overset{Young, s>0}{\leq} (1+s^{-1})\left\|\widetilde{\mathrm{prox}}_{\gamma f_{\xi_t}}(x_t) - \mathrm{prox}_{\gamma f_{\xi_t}}\right\|^2(x_t) + (1+s)\left\|\mathrm{prox}_{\gamma f_{\xi_t}}(x_t) - x_\star\right\|^2$$

$$\leq (1+s^{-1})b + (1+s)\left\|\mathrm{prox}_{\gamma f_{\xi_t}}(x_t) - x_\star\right\|^2.$$

Then proof follows same path as proof Theorem 1 and we get

$$\mathbb{E}\left[\|x_{t+1} - x_\star\|^2\right] \leq (1+s^{-1})b + (1+s)\frac{1}{(1+\gamma\mu)^2}\left(\|x_t - x_\star\|^2 + \gamma^2\sigma_\star^2\right)$$

$$= \frac{1+s}{(1+\gamma\mu)^2}\left(\|x_t - x_\star\|^2 + \left[\gamma^2\sigma_\star^2 + s^{-1}b(1+\gamma\mu)^2\right]\right).$$

azc It only remains to solve the above recursion. Luckily, that is exactly what Fact F.5 does. In particular, we use it with $s_t = \mathbb{E}\left[\|x_t - x_\star\|^2\right]$, $A = \frac{1+s}{(1+\gamma\mu)^2}$ and $B = \frac{(1+s)\left(\gamma^2\sigma_\star^2 + s^{-1}b(1+\gamma\mu)^2\right)}{(1+\gamma\mu)^2}$ to get

$$\mathbb{E}\left[\|x_t - x_\star\|^2\right] \leq A^t\|x_0 - x_\star\|^2 + B\frac{1}{1-A}$$

$$\leq A^t\|x_0 - x_\star\|^2 + B\frac{(1+\gamma\mu)^2}{(1+\gamma\mu)^2 - 1 - s}$$

$$\leq A^t\|x_0 - x_\star\|^2 + \frac{(1+s)\left(\gamma^2\sigma_\star^2 + s^{-1}b(1+\gamma\mu)^2\right)}{(1+\gamma\mu)^2 - 1 - s}$$

$$= \left(\frac{1+s}{(1+\gamma\mu)^2}\right)^t\|x_0 - x_\star\|^2 + \frac{(1+s)\left(\gamma^2\sigma_\star^2 + s^{-1}b(1+\gamma\mu)^2\right)}{\gamma^2\mu^2 + 2\gamma\mu - s}.$$

$\square$

