# OpenReview forum: "Cohort Squeeze: Beyond a Single Communication Round per Cohort in Cross-Device Federated Learning"
_ICLR.cc/2025/Conference — ICLR 2025 Conference Withdrawn Submission_

### Official Review · Reviewer_Qe97 · 2024-10-27

**Soundness:** 2
**Presentation:** 3
**Contribution:** 2
**Rating:** 5
**Confidence:** 2

**Summary:**

The paper presents an innovative method in the domain of federated learning, breaking away from the conventional approach where client cohorts interact with a central server solely once per training cycle. The authors have developed SPPM-AS (Stochastic Proximal Point Method with Arbitrary Sampling), a technique that facilitates additional communication rounds within each cohort, potentially slashing the overall communication expenditure needed for cross-device model training.

Theoretical underpinnings of SPPM-AS are thoroughly examined, with a focus on its convergence characteristics, and are juxtaposed with those of traditional methods. The study delves into the effects of various hyperparameters—including the learning rate and frequency of local communications—on algorithmic performance. Empirical evaluations conducted across both convex (logistic regression) and non-convex (neural network) models substantiate the method's proficiency in lowering communication expenses without compromising accuracy, and in some cases, even enhancing it over current methodologies.

**Strengths:**

1. The research provides a thorough theoretical underpinning to the SPPM-AS method, complete with convergence proofs. This not only bolsters the credibility of the approach but also offers a deeper understanding of its operational dynamics. The paper goes beyond mere theoretical exposition by delivering comprehensive interpretations of the theoretical outcomes, making the material more accessible and applicable for readers.

2. A significant aspect of the paper is its in-depth exploration of diverse sampling strategies, each accompanied by a detailed explanation and analysis. The authors present the sampling variance for each strategy and offer a comparative analysis, highlighting the nuances and implications of choosing one strategy over another. This meticulous examination of sampling strategies enriches the paper's contribution to the field of federated learning.

3. The empirical validation of the theoretical findings is a testament to the practical viability of the SPPM-AS method. Through a series of extensive experiments on both convex and non-convex models, the paper demonstrates the method's robustness and effectiveness in real-world scenarios. These experiments solidify the theoretical claims and showcase the method's potential to be integrated into existing federated learning frameworks, thereby bridging the gap between theory and practice.

**Weaknesses:**

The core novel contributions of this paper are still unclear to me. I appreciate the detailed explanations of the theoretical results and the examples with various concrete sampling strategies. However, the novel algorithm SPPM-AS seems to heavily rely on SPPM, which can already be applied directly to the federated learning setting (Equation 1). I want to understand the technical differences and contributions of SPPM-AS compared to the SPPM algorithm. Please provide a more explicit comparison between SPPM-AS and SPPM, highlighting the key technical differences and innovations.

Moreover, It appears that this paper eliminates the need for the second-order similarity condition in SPPM. How eliminating the second-order similarity condition can be achieved in your proof is of great interest to me.

Last but not least, explain in more detail how the multiple communication rounds within cohorts contribute to the novelty of the approach.

**Questions:**

See weaknesses.

---

### Official Review · Reviewer_sJBf · 2024-11-03

**Soundness:** 2
**Presentation:** 1
**Contribution:** 2
**Rating:** 3
**Confidence:** 3

**Summary:**

Based on SPPM, this paper proposes SPPM-AS, a cross-device federated learning framework that supports arbitrary sampling strategies. The performance of SPPM-AS is evaluated both theoretically and numerically.

**Strengths:**

This paper introduces a new cross-device federated learning framework called SPPM-AS that supports arbitrary sampling strategies. The effectiveness of SPPM-AS is validated through both theoretical analysis and numerical experiments.

**Weaknesses:**

1. The presentation of the paper needs to be improved. For example, it is not easy to follow the paper since a lot of important discussions and results are presented in the appendix.
2. A detailed explanation of Algorithm 1 or SPPM should be provided to improve better reader understanding.
3. The theoretical analysis is based on the strongly convex assumption. Extending the analysis to a more general non-convex setting would strengthen the paper.
4. The comparisons between different sampling methods are based on simplified settings, e.g., $b$ clusters of uniform size $b$, with blocking size and the number of blocks set as 2.
5. The authors only provide experiments on logistic regression using datasets from the LibSVM repository and on CNN with FEMNIST dataset, which are relatively simple. To better demonstrate the performance of SPPM-AS, experiments on more complex datasets (e.g., CIFAR-100, Shakespeare) and tasks (e.g., NLP) are recommended.


Minor:
1. Notations should be explained when they first appear in the paper, e.g., $n$.
2. In line 93, "dashed line" should be corrected to "dashed red line".
3. Abbreviations should be defined upon their first appearance in the paper, e.g., $HP$.

**Questions:**

1. The authors claim that increasing the number of local communication rounds can reduce the total cost. Does this claim hold for all numbers of local communication rounds, or is there a tradeoff between local communication rounds and total cost?
2. The authors state that the stratified sampling optimal clustering is impractical, so they employ a clustering heuristic which is K-means. What are the differences between these two methods?
3. The authors indicate that stratified sampling outperforms nice sampling. Why do they provide the experiment results of CNN under nice sampling rather than stratified sampling?

---

### Official Review · Reviewer_ubzA · 2024-11-04

**Soundness:** 2
**Presentation:** 2
**Contribution:** 2
**Rating:** 3
**Confidence:** 2

**Summary:**

This paper applies the stochastic proximal point method (SPPM) to federated learning. Convergence analysis of SPPM with strongly convex objectives are given, experiments showing that SPPM can reduce the total communication cost compared with FedAvg.

**Strengths:**

- The paper is easy to read in general.

**Weaknesses:**

- In section 2.2, the author(s) discussed some properties of the SPPM-AS, but I cannot find the communication cost analysis of SSPM-AS, which is the most important factor of FL algorithms. Theoretically, how does the total communication cost of SPPM-AS compared with existing FL algorithms such as FedAvg, FedProx, SCAFFOLD, etc.
- Similar to the question above, how is $prox_{\gamma f_{S_t}} ( x_t )$ being solved? There must be some communication between $S_t$ during the optimization, how expensive is the communication?
- Table 1 is not very easy to read. I did not fully get the meaning between 313-323 when I read it for the first time.
- In line 340, how is $\tilde{f}_i$ defined?
- In experiments, how to solve the proximal point problem is kind of vague, what is the local communication cost and how is the local communication cost being controlled in each experiment?
- Federated leaning has been studied many years. The baseline methods in the experiments is limited (FedAvg), the author(s) should include some more recent FL algorithms.

**Questions:**

Please see my comments in the weakness part.

---

### Official Review · Reviewer_7q4p · 2024-11-04

**Soundness:** 2
**Presentation:** 3
**Contribution:** 2
**Rating:** 3
**Confidence:** 4

**Summary:**

This paper presents SPPM-AS, a variant of the stochastic proximal point method that supports various protocols for sampling data. For federated learning, this translates to a federated optimization algorithm that supports various protocols for sampling clients. The method is proven to converge to an $\epsilon$-approximate solution for strongly convex problems, and experiments show improvements compared to classical baselines.

**Strengths:**

1. The sentence-by-sentence writing is clear.
2. All of the proofs seem to be correct.

**Weaknesses:**

1. I don't see any significant theoretical improvement of the proposed algorithms.

    1a. The iteration complexity is $1/\epsilon$ (Line 212), which does not seem to improve upon FedAvg expect possibly in terms of constant factors. This is not too surprising considering that FedProx does not improve upon FedAvg, but there are already works in FL showing that the order of client sampling can improve convergence rate in terms of the $\epsilon$ dependence [1]. Therefore the iteration complexity shown in this paper is not a significant improvement.

    1b. The algorithm which enjoys theoretical guarantees cannot actually be implemented, because the choice of hyperparameters requires knowledge of the global minimum. The iteration complexity (Line 212) requires to choose the stepsize based on $\sigma_{\*,\text{AS}}^2$, which can only be computed with knowledge of $x_*$.

    1c. The only possibility for theoretical improvement is an improvement of constant factors from optimal stratified sampling (Lemma 2), but *optimal stratified sampling cannot be computed without knowledge of the global minimum*.

2. I don't see any significant experimental improvement due to issues with the experimental methodology.

    2a. The experimental evaluation only compares against naive baselines of Local GD and Minibatch GD. There are a huge number of works in FL that try to improve optimization with different client selection strategies, and these works are essentially ignored by this experimental evaluation (see [2] and its references). The authors' claim of $74$% improvement compares against the naive baseline, not against state-of-the-art (or any of the relevant existing work).

    2b. SPPM-SS cannot be implemented with real data. As I pointed out in Weakness #1c, optimal stratified sampling cannot be computed without knowledge of the global minimum. To run experiments, the authors instead use a clustering heuristic that stratifies clients according to features, clustered using K-means. However, it is unclear whether such a clustering procedure can be executed in a real federated learning scenario when client data must remain on-device. Without this, a significant portion of the experimental results (Figures 1, 2, part of 3) only describe an algorithm which cannot be implemented in practice.

    2c. The neural network experiments (Figure 4) may not be a fair comparison between LocalGD and SPPM-NICE. SPPM-NICE uses Adam as a local prox solver, which may not be a reasonable comparison against LocalGD, since LocalGD does not include any update preconditioning (known to be important for neural network training). It would be more appropriate to compare SPPM-NICE against LocalGD when SPPM-NICE uses GD as a local prox solver. An alternative is to compare SPPM-NICE w/ Adam against a local version of Adam, for example FedAdam. Appendix E.4 contains NN experiments with different local solvers (Figure 16), but I don't see exactly how these results related to those in Figure 4. It looks like the choice of local solver can create a gap of about 6\% in train accuracy, and this is described as "all methods perform similarly" (Line 1526), whereas a similar gap between LocalGD vs. SPPM-NICE in Figure 4 is described as "enhanced performance" (Line 513).

3. The paper exaggerates its own contribution and ignores relevant previous work. There are a huge number of works that improve federated optimization with different client selection strategies, which are ignored by this paper in terms of theory, experiments, and general framing (see [2] and its references). Some examples of exaggerated language that I find inappropriate:
- Abstract: "Virtually all FL methods operate in the following manner..." This claim is not accurate; there are many works in FL that use peer-to-peer communication [3], asynchronous communication [4], etc. Further, I fail to see how the proposed algorithms of this paper do not also fall into the category described in the abstract.
- Line 524: "This foundational work showcases a pivotal shift in federated learning strategies". I don't believe that this work departs very far at all from previous work in FL (e.g. [5] and related works). In my opinion, this kind of self-aggrandizing is not appropriate for a scientific publication.

4. The message of the paper is not totally coherent. The abstract talks about "cohort squeeze" and novel communication principles, but most of the paper actually deals with client selection strategies within the standard intermittent communication structure. The experiments discuss local vs. global communication (Section 3.6), which seems to be the connection to the "cohort squeeze" of the title and abstract, but this section makes up a very small part of the paper's technical content. Perhaps I have missed a connection between the content of the abstract and the content of the main text.

[1] Cho, Yae Jee, et al. "On the convergence of federated averaging with cyclic client participation." International Conference on Machine Learning. PMLR, 2023.

[2] Fu, Lei, et al. "Client selection in federated learning: Principles, challenges, and opportunities." IEEE Internet of Things Journal (2023).

[3] Beltrán, Enrique Tomás Martínez, et al. "Decentralized federated learning: Fundamentals, state of the art, frameworks, trends, and challenges." IEEE Communications Surveys & Tutorials (2023).

[4] Xu, Chenhao, et al. "Asynchronous federated learning on heterogeneous devices: A survey." Computer Science Review 50 (2023): 100595.

[5] Grudzień, Michał, Grigory Malinovsky, and Peter Richtárik. "Improving accelerated federated learning with compression and importance sampling." arXiv preprint arXiv:2306.03240 (2023).

**Questions:**

1. Do any of the proposed algorithm variations achieve any theoretical speedup compared to Local GD with i.i.d. client sampling, beyond an improvement in constant factors?
2. Is there any way to execute optimal stratified sampling in practice?
3. How does your algorithm compare experimentally against baselines that use client selection strategies besides NICE sampling, e.g. Power-of-Choice [6]?
4. In the neural network experiments (Figure 4), how does SPPM-NICE compare against LocalGD when SPPM-NICE uses GD as a local prox solver instead of Adam?

[6] Jee Cho, Y., Wang, J. &amp; Joshi, G.. (2022).  Towards Understanding Biased Client Selection in Federated Learning . Proceedings of The 25th International Conference on Artificial Intelligence and Statistics, in Proceedings of Machine Learning Research.

---

> ### Comment · Reviewer_7q4p · 2024-11-26
>
> Just want to follow up on my review. Do the authors plan to participate in the discussion period? If so, I am happy to discuss my concerns.

---

### Official Review · Reviewer_HpK8 · 2024-11-18

**Soundness:** 2
**Presentation:** 3
**Contribution:** 2
**Rating:** 3
**Confidence:** 4

**Summary:**

This paper studied the problem of whether one can change the conventional operation in FL, where a cohort of client devices can be involved in multiple rounds of communication with the server. The authors proposed a variant of the stochastic proximal point method (SPPM-AS), which supports a large collection of client sampling procedures to lead to further gains compared to classical client selection approaches. The authors further conducted experiments to verify the performance of the proposed SPPM-AS algorithm.

**Strengths:**

1. The authors proposed a new variant of federated learning algorithms to further improve the communication efficiency in federated learning.

2. The experiments in this paper is comprehensive.

**Weaknesses:**

1. The federated learning setting considered in this paper are limited. Specifically, this paper assume that the objective function is strongly convex (Assumption 2.2). This not only significantly simplifies the theoretical analysis to achieve stronger convergence results, but also is not of practical interests since most ML models are non-convex.

2. The algorithm design is only a minor variation of the existing algorithmic framework in federated learning, which has been well-explored. Algorithmic ideas such as proximal point optimization, various types of client sampling, and multiple local updates for each cohort are not new and have been considered in the literature. It's unclear what are the major novelty in this paper.

3. As a consequence of the above factors, the theoretical contributions of this paper are marginal, since most of the proof details in Appendix F are quite standard.

**Questions:**

1. Could theoretical convergence performance3 analysis of the proposed method be generalized to non-convex settings?

2. If yes to the above question, what are the major challenges in the theoretical analysis and how to overcome them?

---

### Note · Authors · 2024-11-30

I have read and agree with the venue's withdrawal policy on behalf of myself and my co-authors.